# MDG625: A daily high-resolution meteorological dataset derived by geopotential-guided attention network in Asia (1940-2023)

Zijiang Song[1,2], Zhixiang Cheng[1,2], Yuying Li[1,2], Shanshan Yu[1,2], Xiaowen Zhang[1,2], Lina Yuan[3,1,2], and Min Liu[1,2]

[1]Key Laboratory of Geographic Information Science, Ministry of Education, School of Geographic Sciences, East China Normal University, Shanghai 200241, China

[2]Key Laboratory of Spatial-temporal Big Data Analysis and Application of Natural Resources in Megacities, Ministry of Natural Resources, Shanghai, 200241, China

[3]School of Geospatial Artificial Intelligence, East China Normal University, Shanghai 200241, China

**Correspondence:** Lina Yuan (lnyuan@geoai.ecnu.edu.cn) and Min Liu (mliu@geo.ecnu.edu.cn)

**Abstract.** The long-term and reliable meteorological reanalysis dataset with high spatial-temporal resolution is crucial for various hydrological and meteorological applications, especially in regions or periods with scarce in situ observations and with limited open-access data. Based on the fifth-generation reanalysis dataset (ERA5, produced by the European Centre for Medium-Range Weather Forecasts, 0.25°×0.25°, since 1940) and CLDAS (China Meteorological Administration Land Data Assimilation System, 0.0625°×0.0625°, since 2008), we proposed a novel downscaling method Geopotential-guided Attention Network (GeoAN) leveraging the high spatial resolution of CLDAS and the extended historical coverage of ERA5 and produced the daily multi-variable (2m temperature, surface pressure, and 10m wind speed) meteorological dataset MDG625. MDG625 (0.0625° Meteorological Dataset derived by GeoAN) covers most of Asia from 0.125° S to 64.875° N and 60.125° E to 160.125° E since 1940. Compared with other downscaling methods, GeoAN shows better performance with the $R^2$ (2m temperature, surface pressure, and 10m wind speed reached 0.990, 0.998, and 0.781, respectively). MDG625 demonstrates superior continuity and consistency from both spatial and temporal perspectives. We anticipate that this GeoAN method and this dataset MDG625 will aid in climate studies of Asia and will contribute to improving the accuracy of reanalysis products from the 1940s. The MDG625 dataset (Song et al., 2024) is presented at https://doi.org/10.57760/sciencedb.17408 and the code can be found at https://github.com/songzijiang/GeoAN.

# 1 Introduction

As temperatures rise and extremes become more frequent, weather-related data analysis is becoming increasingly important (Berrang-Ford et al., 2011; Dietz et al., 2020; Taylor et al., 2013; Karl and Trenberth, 2003). Spatial resolution is crucial for geographic datasets. However, the distribution of in-situ stations is too sparse to produce a high-quality reanalysis dataset, especially for decades ago. For getting a higher resolution reanalysis dataset, downscaling is widely used in geoprocessing (Atkinson, 2013), especially in climate-related fields (Wang et al., 2021; Vogel et al., 2023; Tefera et al., 2024; Sun et al., 2024). The meteorological reanalysis dataset, which is obtained from in situ and remote sensing measurements, is important for agriculture, extreme weather forecasts, etc. Higher resolution of these data can better guide life and production. He et al. (2020) produced a meteorological dataset with a spatial resolution of $0.1°$ from 1979 in China. In this paper, the China Meteorological Forcing Dataset was proposed by fusing remote sensing products, reanalysis datasets, and in-situ station data. The most significant contribution of this work was using a larger number of stations to improve the dataset. A long-term gridded daily meteorological dataset for northwestern North America was proposed by Werner et al. (2019). The authors produced a dataset for training statistical downscaling schemes in Canada. Similarly, Bonanno et al. (2019) proposed the high-resolution meteorological dataset named MERIDA in Italian. MERIDA was produced by dynamical downscaling from the fifth-generation reanalysis (ERA5) dataset for the global climate and weather using WRF. The resolution of the image is fundamentally limited by the optical constraints of the imaging components, and high-resolution reanalysis data is expensive to produce. Moreover, obtaining detailed historical data at high resolution poses a significant challenge, especially since historical observations were limited in number. High-quality and high-resolution data is necessary for various studies, to solve the contradiction, low-resolution (LR) data products being used to downscale into high-resolution (HR, also called as ground truth) are widely used (Hu et al.,

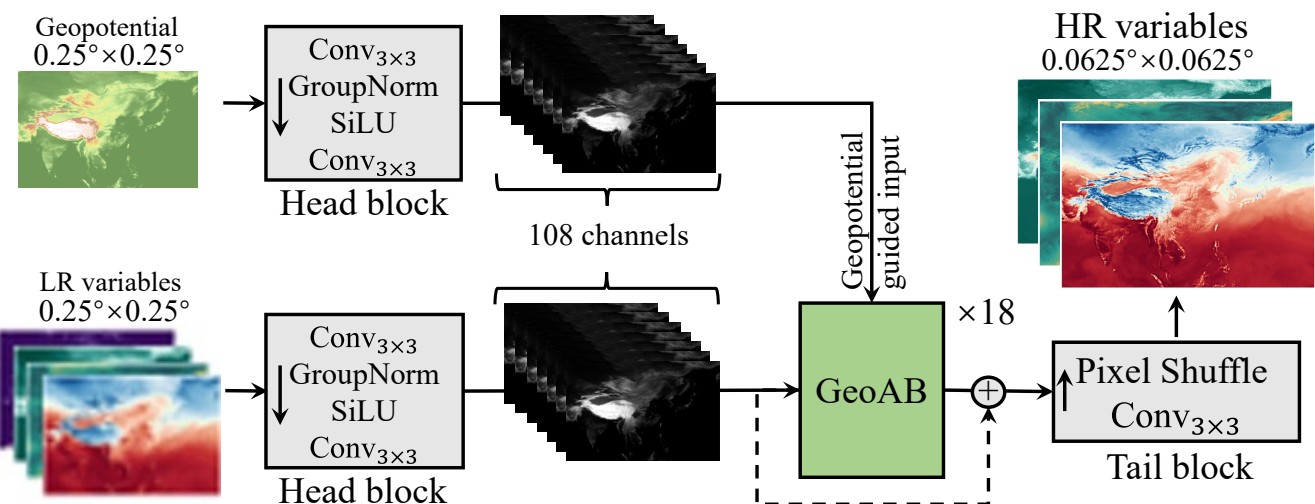

**Figure 1.** Sketch of the GeoAN. LR and HR denoted the low-resolution and high-resolution, respectively. The head block contains one group norm, one activation function, and two convolutions, which are abbreviated by $\mathrm{Conv}_{3\times3}$ meaning the kernel size is $3 \times 3$. SiLU is adopted as the activation function. The results of the two blocks of head in the diagram have the same channels of 108. GeoAB, which is repeated 18 times constricted by the hardware and the data amount, is the attention block for extracting deep information using geopotential. The pixel shuffle operation is performed after the convolution in the tail block to produce the high-resolution variables. Note that, the order of execution in each grey block (i.e., head and tail blocks) is along the arrows in the box.

2023; Zhong et al., 2023). The mainly used downscaling methods are categorized into statistical downscaling and dynamical
downscaling. While the existing downscaling methods could produce high-resolution results, the results are unsatisfactory and unable to reconstruct the absent details and use information from the long historical samples (Murphy, 1999), meteorological data exhibit a high degree of historical similarity and long-term historical records can effectively guide the generation of downscaled data. Dynamical downscaling methods are usually based on Regional Climate Models (RCMs) with the initial fields produced by Global Climate Models (GCMs). Although RCMs offer higher resolution than GCMs, the comprehension
ability to accurately represent the real world is not enough. It leads to a considerable bias (Teutschbein and Seibert, 2012). The method challenges in developing precise simulation equations, which often struggle to capture the complexity of natural systems. In another opinion, the computational cost of RCMs is huge, and it is an obstacle to producing a wider range of results for each calculation (Giorgi and Gutowski Jr, 2015; Di Luca et al., 2015). Compared with dynamical downscaling, statistical downscaling uses the mapping relationship between high-resolution and low-resolution from historical datasets to produce
future datasets. The computational cost and bias of statistical downscaling are lower than dynamical downscaling methods.

Deep learning is a statistical method to build the bridge between input and output. Since Vaswani et al. (2017) proposed the transformer network, the ability of deep learning to harvest shadow information has gained a step. After that, the transformer block is widely used in diverse tasks including Super-Resolution (SR) (Liang et al., 2021; Zhang et al., 2022; Song and Zhong, 2022). Liang et al. (2021) proposed SwinIR and achieved impressive results in the SR task and be considered the benchmark

for the SR task. The core algorithm of SwinIR is using no overlap windows to split the input feature to calculate the attention relationship inner each window and shift the windows by the step of the half-width of the windows. Song and Zhong (2022) proposed a novel network to harvest long-range information from global instead of inner the window. The experimental results on SR benchmarks (Bevilacqua et al., 2012; Martin et al., 2001; Huang et al., 2015; Matsui et al., 2017) show this strategy can achieve better results, such PSNR and SSIM. Super-resolution tasks share similarities with geographic downscaling tasks. Applying deep learning-based super-resolution techniques to downscale geographical data can successfully overcome issues associated with traditional downscaling approaches, such as high bias, regional sensitivity, and high computational expenses. Deep learning techniques employ multiple layers to link low-resolution inputs with high-resolution outputs while demonstrating robustness to sensitivity issues. After training, the computational expense remains minimal during deployment. The deep learning approach can easily accommodate a wide range of applications. Shen et al. (2023) proposed a near-surface air temperature downscaling network SNCA-CLDASSD. In their framework, Shen et al. used two attention blocks designed to downscale the input through a technique known as Cross-Attention, which is based on principles from Light-CLDASSD. However, only near-surface air temperature is considered in this work and the network was built on CLDAS (China Meteorological Administration Land Data Assimilation System), which does not support multi-year data coverage. Liu et al. (2023) used the terrain to guide the deep learning network for the downscaling task called terrain-guided attention network (TGAN) in Southwest China. TGAN used the digital elevation model (DEM) to build high-resolution temperature (at 2 meters) results. The application of TGAN started from 2018, and it is not applicable in the historical situations. Zhong et al. (2023) proposed a transformer-based learning method Uformer, which directly adds topography data, to achieve high-resolution meteorological variables in inner Mongolia province, China. Although topography data can help rebuild the high-resolution, directly adding into the input low-resolution will lose the characters of topography. The current advanced deep learning techniques for meteorological downscaling predominantly employ attention-based architectures. Nevertheless, current approaches concentrate on just one or two meteorological variables. Variables are interconnected, and deep learning can process multiple variables simultaneously. Calculating several not only reduces computational demands but also boosts model effectiveness. Most crucially, no existing models can analyze extensive historical data covering various variables over a long period.

In this paper, we propose a novel attention-based network called the Geopotential-guided Attention Network (GeoAN), designed for downscaling meteorological variables such as temperature at 2 meters (T2m), surface pressure (PRS), and wind speed at 10 meters (WS10m) from a coarser resolution of $0.25°$ to a finer resolution of to $0.0625°$. The architecture of GeoAN is illustrated in Fig. 1. The proposed GeoAN is guided by the geopotential, which makes the model learn information in a targeted manner instead of unguided or random learning processes. The low-resolution input of the variables is sourced from ERA5, provided by the European Centre for Medium-Range Weather Forecasts (ECMWF), with data ranging from 1940 to the present. The target data for the downscaling algorithm is derived from CLDAS (Shi et al., 2014; Sun et al., 2020; Shi et al., 2011), which provides high-quality, high-resolution daily data. However, CLDAS data is only available for the years after 2008, creating a gap in historical records. To address this limitation, we employed deep learning networks to establish a robust mapping relationship between ERA5 and CLDAS datasets. By training the GeoAN model on the part overlapping period (2020 onwards), we generated a consistent and accurate historical meteorological dataset extending back to 1940. This

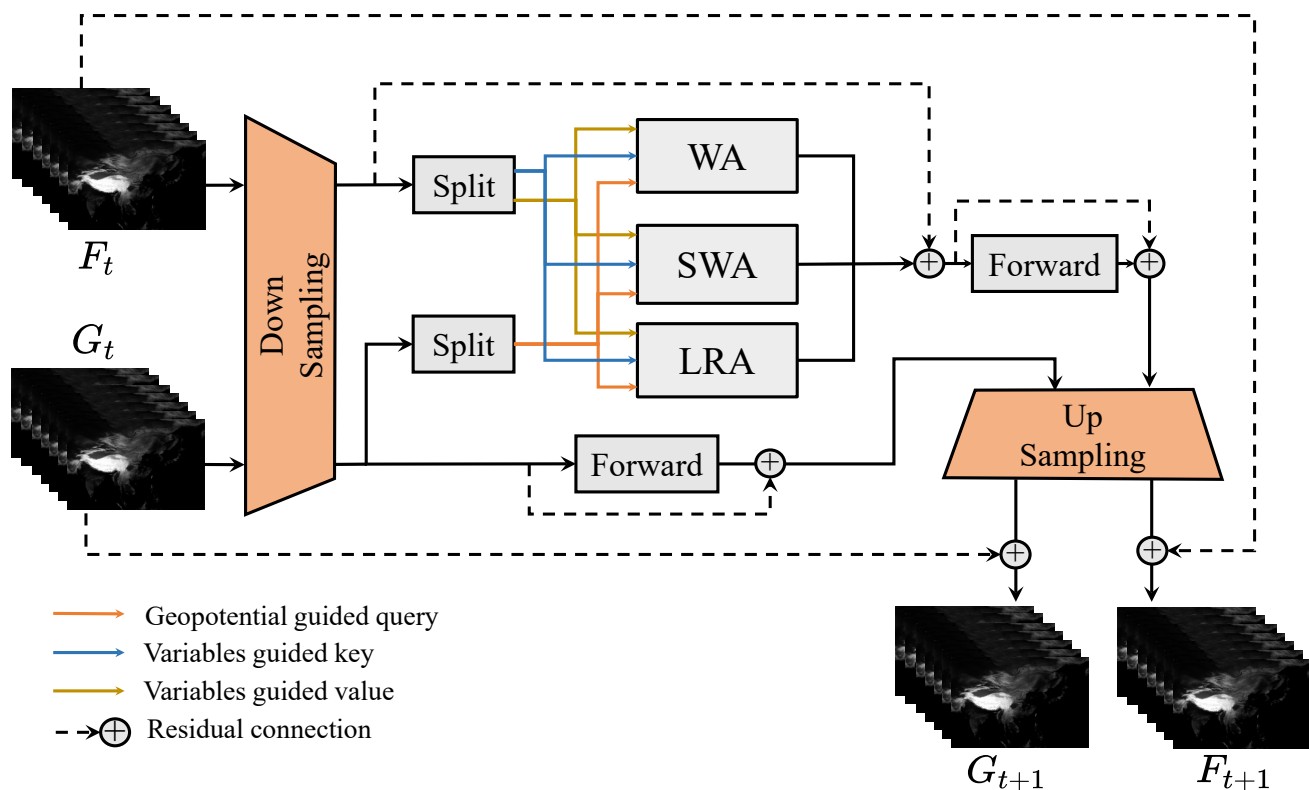

**Figure 2.** Sketch of the GeoAB, which repeated 18 times in GeoAN. GeoAB is the attention block to extract deep information. The query information of GeoAN is harvested from geopotential and the key and value are made from variable features. To make the loops, the outputs of the $t^{th}$ GeoAB, i.e., $F_{t+1}$ and $G_{t+1}$, are treated as the input of the $(t+1)^{th}$ GeoAB.

newly created dataset, referred to as MDG625, effectively fills the gap in CLDAS data prior to 2008 while also enhancing the spatial resolution of ERA5 data. The development of MDG625 is particularly valuable for various applications, including climate change studies and the analysis of extreme weather events. The increased spatial and temporal resolution provided by MDG625 allows researchers to perform more detailed analyses, offering insights into long-term climate trends and high-resolution weather patterns that were previously challenging.

## 2   Data and methods

### 2.1   Data

The study area spans most of Asia (latitudes from $0.125°$ S to $64.875°$ N and longitudes from $60.125°$ E to $160.125°$ E), including China, Japan, India, etc. ERA5 is the fifth generation ECMWF reanalysis, provided by the ECMWF and used widely (Muñoz-Sabater et al., 2021; Hersbach et al., 2020; Jiang et al., 2021; Olauson, 2018; Cucchi et al., 2020), for the global

climate and weather. ECMWF is a premier international organization, considered advanced in numerical weather prediction (NWP) models. The variables of PRS and T2m used in this study are directly sourced from the ERA5 dataset, while WS10m is calculated from U and V components of the wind at 10m. CLDAS, which uses multigrid variational analysis and multi-source precipitation fusion, is a reanalysis production provided by the China Meteorological Administration (CMA). The high-resolution data in CLDAS appears to be more accurate and reliable compared to other datasets. In this paper, ERA5 is

used as the low-resolution image (LR), and CLDAS is treated as the high-resolution image (HR, i.e., ground truth) to train the proposed model. The output from the downscaling network is called super-resolution images (SR).

There are four meteorological variables, temperature at 2m, pressure at the surface, wind speed at 10m, and daily total precipitation (TP) considered in GeoAN. Considering it is hard to process the downscale of TP, only three other variables are produced by GeoAN in MDG625. The period of dataset used to train the network spans from 2020 to 2022, while the validation

dataset covers the entire year of 2023. It is important to mention that all times are referenced in Coordinated Universal Time (UTC). The spatial resolution of ERA5 and CLDAS are $0.25°$ and $0.0625°$ respectively. The temporal resolution of these two datasets is calculated to one day, which is calculated by the mean of PRS (hPa), T2m (K), WS10m ($\mathrm{m \cdot s^{-1}}$) and the sum of TP (mm) over the whole day respectively using the corresponding hourly data. The region is defined by CLDAS boundaries, i.e. latitudes from $0.125°$S to $64.875°$N, and the range of longitudes is from $60.125°$E to $160.125°$E. However, since the grid

systems of ERA5 and CLDAS do not align perfectly, the spatial extent of ERA5 is slightly broader, covering latitudes from $0.25°$S to $65°$N and longitudes from $60°$E to $160.25°$E.

## 2.2 Geopotential-guided attention network

Geopotential ($\mathrm{m^2 \cdot s^{-2}}$) is the gravitational potential energy of a unit mass. Geopotential can reflect the elevation, latitude, pressure, etc. The value of geopotential used in this paper is obtained from the ERA5 dataset. Using geopotential to guide the

115 attention calculation for downscaling can gain geographic semantic information, which is lacking in common deep learning networks.

As shown in Fig. 2, geopotential-guided attention is realized by the Geopotential-guided Attention Block (GeoAB), which is the core unit of the GeoAN. The window attention (WA), shifted window attention (SWA), and long rang attention (LRA) are constructed from Song and Zhong (2022) and Song et al. (2022). The concepts of query, key, and value were used in

transformer block Vaswani et al. (2017) to excavate the effects of attention. Unlike the original design where the query is derived from input features, our approach generates it from geopotential. However, the key and value remain sourced from input features, consistent with the traditional method. For ease of understanding, normalization, residual operation, and other detailed parts are not listed in the formulas incidentally. The formulas are defined as follows, where $F_t$ and $G_t$ denoted the deep features of meteorological variables and geopotential at $t^{th}$ loop respectively:

$$F_{t+1} = \mathcal{F}(\mathcal{A}(G_t, F_t)),\qquad(1)$$

$$G_{t+1} = \mathcal{F}(G_t).\qquad(2)$$

where $\mathcal{F}(\cdot)$ and $\mathcal{A}(\cdot)$ denoted the forward and attention parts respectively, it is important to note that none of the forward components shared their parameters. Additionally, both the WA and SWA were upgraded from Swin Transformer (Liu et al., 2021) to Swin Transformer V2 (Liu et al., 2022) comparing Song and Zhong (2022). The network architecture is described in Fig. 1. Based on prior experience, GeoAB is executed 18 times to gather more geographic data, as shown in Eq. 1, Eq. 2, and Fig. 2, the definition of the network architecture is described as follows:

$$\text{SR} = \mathcal{T}(\text{GeoAB}^{18}[\mathcal{H}(LR), \mathcal{H}(G)]), \tag{3}$$

where $LR$, $G$, and $SR$ denoted low-resolution variables, geopotential, and produced high-resolution variables, respectively. $\mathcal{H}(\cdot)$, $\mathcal{T}(\cdot)$, and $\text{GeoAB}^{\text{k}}[\cdot, \cdot]$ denoted head block, tail block, and GeoAB block, respectively. The GeoAB block is repeated for $k$ times.

The batch size of the training step was 5, which is an unbalanced GPU distribution for 3 NVIDIA RTX 6000 Ada Generation (48G), considering the GPU memory limitation. There were 6 days of high-resolution data missing from 2020 to 2022. Therefore, the training set consisted of 1,090 LR and SR pairs, which is derived from adding 366, 365, 365, and subtracting 6. The learning rate was set to $10^{-4}$ and reduced by half at epochs 20, 40, 60, 80, 90 and 95. The network was trained for 100 epochs from the pre-trained models. Considering differences among the lines of latitude, the latitude-weighted loss was chosen to be the loss function, and the distortion of geographical coordinates with changes in latitude is fully taken into account (Bi et al., 2023; Rasp et al., 2020). The loss function is defined as follows:

$$loss = \frac{\sum_{i=1}^{\text{H}} \sum_{j=1}^{\text{W}} \sum_{c=1}^{\text{C}} a_i \times |\text{HR}_{i,j,c} - \text{SR}_{i,j,c}|}{\text{H} \times \text{W} \times \text{C}}, \tag{4}$$

where H, W, and C are 1040, 1600, and 4, respectively (1040 and 1600 represent the pixel counts along latitude and longitude, and 4 represent WS10m, T2m, PRS, and TP). $\text{HR}_{i,j,c}$ and $\text{SR}_{i,j,c}$ is the value at position of $(i, j)$ of channel $c$ in HR variables and SR variables. The $a_i$ is latitude weight defined as:

$$a_i = \text{H} \cdot \frac{\cos\theta_i}{\sum_{i=1}^{\text{H}} \cos\theta_i}, \tag{5}$$

where $\theta_i$ is the latitude of the $i^{th}$ line in the map of the variables in the form of $1040 \times 1600 \times 4$. For calculation purposes, the latitudes range is offset to 0 - 65°N (i.e., $0 <= \theta_i < 65\frac{\pi}{180}$) replacing 0.125°S - 64.875°N.

## 3 Performance

### 3.1 Quantitative comparison

This section details experiments to evaluate the performance of GeoAN. For comparison, the classic algorithm bilinear interpolation, widely used in downscaling, is included. Additionally, two deep learning methods, U-Net (Ronneberger et al., 2015) and SwinIR (Liang et al., 2021), were employed for comparative analysis. The source code for both networks was obtained from their perspective GitHub repositories. To ensure a fair comparison, the U-Net architecture was modified for

**Table 1.** A comparison of our proposed GeoAN with other downscaling methods. The bigger value stands for better performance, and the value in bold indicates the best performance in each metric. Considering the suitability of the downscaling task, PSNR, SSIM, and $R^2$ are chosen. All results are produced by the same environment and super parameters.

| Methods | Variables | PSNR (dB) ↑ | SSIM ↑ | $R^2$ ↑ |
|---------|-----------|-------------|--------|---------|
| **Bilinear** | **T2m** | 27.920 | 0.900 | 0.939 |
|  | **WS10m** | 21.271 | 0.747 | 0.582 |
|  | **PRS** | 33.392 | 0.902 | 0.965 |
| **U-net (Evol.)** | **T2m** | **35.471** | 0.969 | **0.991** |
|  | **WS10m** | 25.556 | 0.845 | 0.780 |
|  | **PRS** | 40.008 | 0.969 | 0.990 |
| **SwinIR** | **T2m** | 34.042 | 0.956 | 0.988 |
|  | **WS10m** | 24.452 | 0.825 | 0.745 |
|  | **PRS** | 37.435 | 0.943 | 0.978 |
| **GeoAN (Ours)** | **T2m** | 35.054 | **0.983** | 0.990 |
|  | **WS10m** | **25.599** | **0.859** | **0.781** |
|  | **PRS** | **47.251** | **0.996** | **0.998** |

the downscaling task, resulting in a customized version referred to as U-Net Evolution (U-Net Evol.). The original U-Net implementation is available at https://github.com/milesial/Pytorch-UNet, while the SwinIR code can be accessed at https://github.com/JingyunLiang/SwinIR. To maintain consistency, all deep learning models were configured with equivalent pa-
160 rameters or computational complexity, and they were trained for 100 epochs using identical hyperparameters under the same environmental conditions.

As shown in Tab. 1, PSNR (Peak Signal-to-Noise Ratio), SSIM (Structural Similarity Index), and $R^2$ (Coefficient of Determination) are considered to evaluate the performance of the methods. PSNR and SSIM are the most commonly used metrics for measuring super-resolution algorithms. Compared to RMSE (Root Mean Squared Error), $R^2$ or other numerical metrics
only calculating the individual value of each pixel, a more holistic and detailed assessment is considered in PSNR and SSIM. A PSNR greater than 25dB is acceptable and greater than 30dB is considered a good result. In most metrics, GeoAN produces better results than others, however, in the T2m comparison, U-net (Evol.) got a higher result, further analyses about this part will be discussed in the appendix. It's important to note that SwinIR's results are worse to those of U-net (Evol.) This discrepancy can likely be attributed to the limited training process, which consisted of only 100 epochs in these experiments due to
GPU constraints. This training process may be insufficient for attention-based models like SwinIR or GeoAN to reach their full potential. However, in this situation, GeoAN could outperform the other methods.

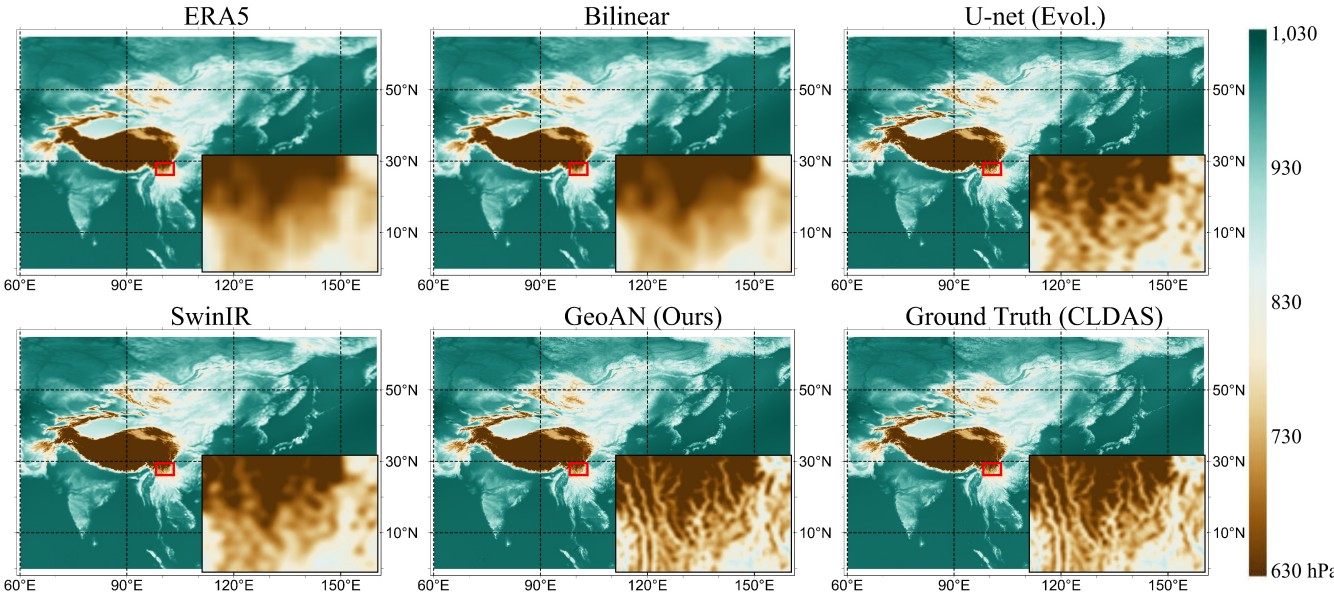

**Figure 3.** Pressure visual results of GeoAN and other downscaling algorithms on the 1st of November 2023. GroundTruth means the target high-resolution data (i.e., CLDAS), and ERA5 is the original low-resolution data. GeoAN is the deep learning method we proposed in this paper. The picture in the lower right corner of each subgraph is the detailed picture of the target area (i.e., red rectangle) respectively.

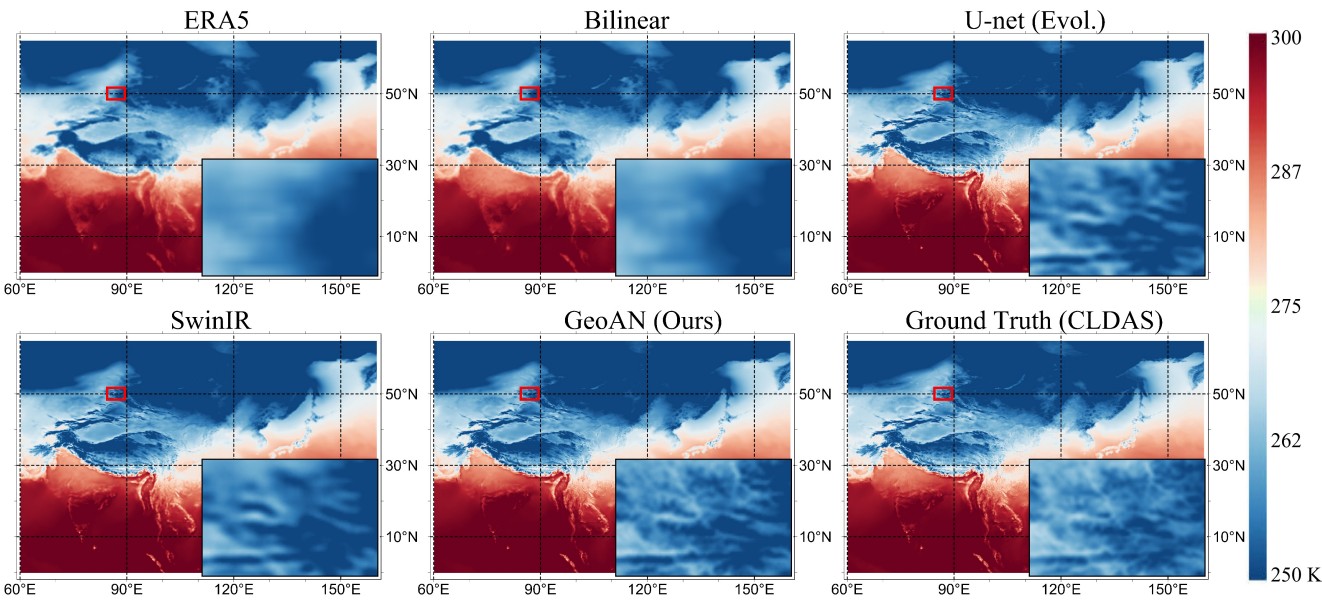

**Figure 4.** Temperature visual results of GeoAN and other downscaling algorithms on the 1st of January 2023.

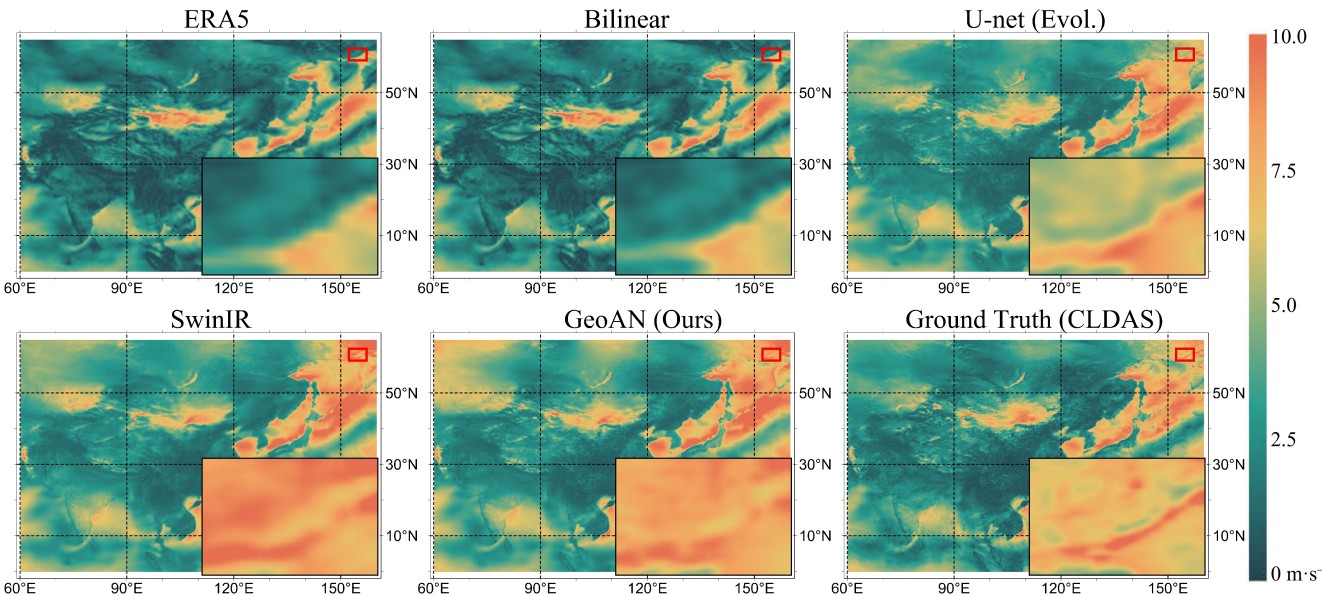

**Figure 5.** Wind speed visual results of GeoAN and other downscaling algorithms on the 1st of November 2023.

## 3.2 Visual comparison

Although GeoAN achieved superior performance in most cases in Tab. 1, a qualitative assessment is also necessary for a direct evaluation. Visual results of PRS, T2m, WS10m are shown in Fig. 3, Fig. 4, and Fig. 5, respectively. We compare the 1st of each two months in 2023 (i.e., January, March, May, July, September, and November) and choose one day to display for each variable. As shown in the figures, GeoAN can achieve the best results among all the compared algorithms. Especially, for extracting details, GeoAN has an excellent performance. Thanks to geopotential-based attention guidance and training with historical data, the neural network captures sufficient geographical semantics. Even distorted parts can be effectively reconstructed through GeoAN.

## 4 Produced dataset

### 4.1 Historical meteorological data

The CLDAS has been in operation from 2008 up to today. Its development depends on observational data, which was limited in China prior to the 2000s (Tie et al., 2022). As high-resolution historical meteorological data is difficult to obtain, we employed our developed model, GeoAN, which is guided by geopotential, to generate a dataset known as MDG625 (Meteorological Dataset with $0.0625°$ resolution created by GeoAN) for the study area beginning in 1940. MDG625 is valuable for historical meteorological studies in relevant areas. The comparison between similar datasets is in Tab. 2. The resolution of ERA5 and GLDAS is too low for various regional studies. The CLDAS dataset from CMA lacks sufficient temporal coverage for long-

**Table 2.** A comparison of different datasets.

| Datasets | Time period | Spatial resolution | Derived from | Sources |
|---|---|---|---|---|
| ERA5 | 1940 - present | 0.25° × 0.25° | Reanalysis method | ECMWF |
| CLDAS | 2008 - present | 0.0625° × 0.0625° | Reanalysis method | CMA |
| GLDAS | 1948 - present | 0.25° × 0.25° | Reanalysis method | NASA |
| MDG625 | 1940 - present | 0.0625° × 0.0625° | Deep learning method | Ours |

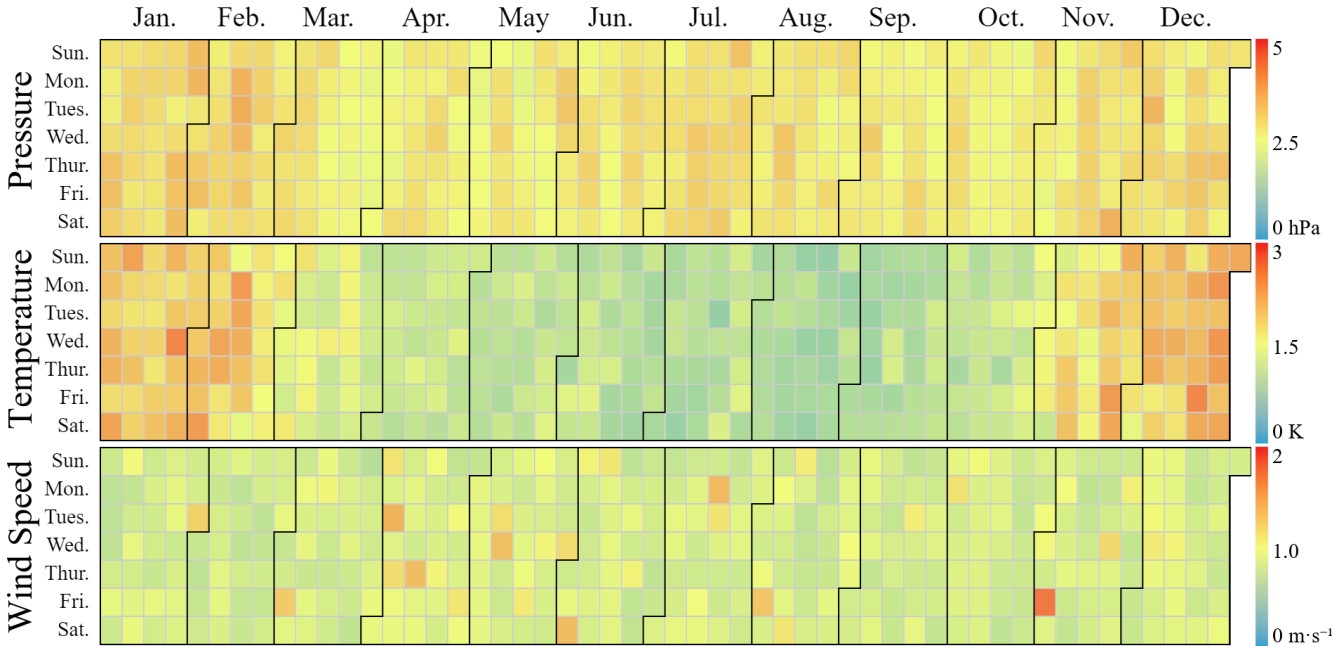

**Figure 6.** The daily average RMSE of MDG625 in 2023. From top to bottom are pressure (hPa), temperature (K) at 2m, and wind speed (m·s⁻¹) at 10m, respectively. The RMSE at a single day is calculated from the daily pixels.

term studies. The key distinction between MDG625 and other models is its reliance on deep learning rather than traditional numerical methods.

Note that, there are two days in ERA5 with anomalies: '1965-11-29' and '2008-7-6'. The first day of MDG625 is '1940-1-1' and the index of this day is assigned an index value of 0. The index of each day represents the number of days elapsed since the 1st of January 1940. Additionally, a more extensive spatial dataset can be generated by GeoAN, however considering the pattern used in training steps, only the data in the study area is provided in MDG625.

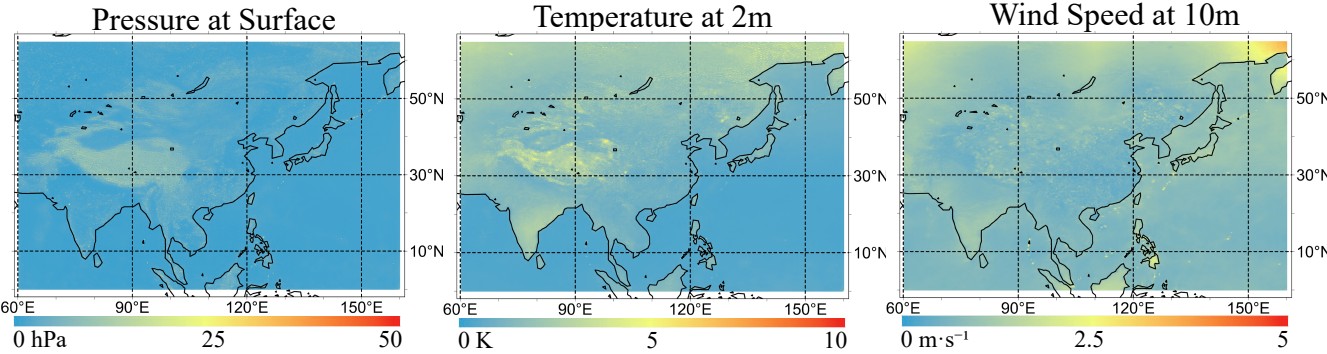

**Figure 7.** The RMSE map of three meteorological variables (PRS, T2m, and WS10m) between MDG625 and the ground truth. The RMSE is calculated from the whole year daily in 2023. Blue represents a smaller error and red represents the bad results.

## 4.2 Error distribution

Considering the period of CLDAS, and the data from 2020 to 2022 are used in the training stage, the results of error distribution are calculated in 2023. The RMSE of the 2m temperature, surface pressure, and 10m wind speed are 1.40 K, 2.76 hPa, and $0.89 \, \mathrm{m \cdot s^{-1}}$ respectively. To further evaluate the quality of MDG625 from the perspective of temporally and spatially, the error distributions of the variables (PRS, T2m, and WS10m) are analyzed in Fig. 6 and Fig. 7. As illustrated in Fig. 6, the variables of T2m in winter do not meet expectations, while other variables demonstrate satisfactory performance. However, although in the worst month of T2m (i.e., January), the difference to ground truth is approximately only 3K, which is deemed acceptable. The average RMSE of T2m is around 1K for the whole year. For PRS and WS10m, stable good performance throughout the whole year. This could be due to seasonal fluctuations in temperature, which challenges a statistical model's ability to produce accurate results when no additional season or date information is provided. One thing to note, there are 9 days missing in CLDAS by the time of the program run. The RMSE of the missing data is calculated by the mean of the nearest data that are not missing before and after the missing date.

In terms of the spatial distribution as shown in Fig. 7. PRS and T2m show better results in marine than in the mainland. In contrast, WS10m results on the mainland are better. For T2m, the results of the mainland performed worse than the marine, which may be caused by the specific heat capacity of water being higher than the land (e.g., soil, sand, etc.) Temperature variations over the oceans are lower in magnitude than over the continents, and the deep learning method is better at learning the patterns of small changes. For the other two variables, the same reason caused the error distribution. Regardless of which variables are considered, the outcomes in coastal plain regions (e.g., such as parts of China's eastern seaboard) tend to be more favorable compared to other areas. Finally, despite some areas showing less-than-satisfactory outcomes, the error levels remain within an acceptable range, enabling the dataset to be effectively utilized for diverse analyses.

## 4.3 Limitations

Although the proposed GeoAN demonstrated satisfactory accuracy in spatial downscaling and the resulting MDG625 is valuable for a wide range of applications, this work still has several limitations. Firstly, constrained by the computational resources and GPU memory limitations, we could not employ a larger model or experiment with a more extensive set of training samples. The generated MDG625 dataset (0.0625°/daily Meteorological Dataset derived by GeoAN) currently covers most of Asia. Higher spatiotemporal resolutions (e.g., hourly) with broader geographical coverage could be explored using the pro-

posed GeoAN framework in future research, as the development of a global high-resolution historical meteorological dataset would be highly valuable. Additionally, precipitation poses a considerable challenge for statistical models, particularly in extreme cases (Zou et al., 2024; Sachindra et al., 2018; Xu et al., 2015). The MDG625 dataset includes three meteorological variables (PRS, T2M, and WS10m) but lacks precipitation data. This exclusion is mainly attributed to the low correlation of total precipitation (TP) between the ERA5 and CLDAS datasets, which causes the GeoAN model to face significant challenges

in accurately reconstructing the spatial structure of precipitation. Addressing this limitation remains a critical area for future improvement. Inspired by the work of (Zou et al., 2024), more interpretable and comprehensible algorithms could be integrated into the GeoAN framework in the future. Furthermore, comparative studies with other models could be conducted to further evaluate and benchmark the performance of our approach.

## 5 Conclusions and discussions

Given the scarcity of long-term, high-resolution historical meteorological data in Asia, the MDG625 dataset (a daily Meteorological Dataset with a resolution of $0.0625°$ containing three variables: 2m temperature, surface pressure, and 10m wind speed) provides a valuable solution. This dataset was produced using a deep geographic coupling attention network called the Geopotential-guided Attention Network (GeoAN), which operates within an acceptable error margin. The GeoAN could directly learn geopotential relationships, which are closely related to meteorological variables. This downscaling strategy en-

hances the network's ability to effectively capture geographic information, producing reliable results. Experimental results have demonstrated the superior performance of the GeoAN framework, and the generated MDG625 is expected to contribute to climate studies in Asia significantly. Additionally, our study can potentially improve the accuracy of reanalysis products from the 1940s onward. In future work, we plan to integrate remote sensing data and gauged precipitation to explore the precipitation downscaling methods using alternative geographic principles and related variables. These findings highlight the potential of

deep learning methods coupled with geographic mechanisms to address various geographic challenges.

*Code and data availability.* The ERA5 data of ECMWF can be found at https://cds.climate.copernicus.eu. The high-resolution data, CLDAS, is provided by CMA at https://data.cma.cn. An education and research account is required to acquire the CLDAS data, this requirement is set by the CMA. The code and the generated dataset MDG625 (Song et al., 2024) can be found in the GitHub repository: https://github.com/

songzijiang/GeoAN and ScienceDB repository: https://doi.org/10.57760/sciencedb.17408. Considering CLDAS is not public, and GeoAN was trained using CLDAS, the data of MDG625 for 2017-2023 are not offered in the repository.

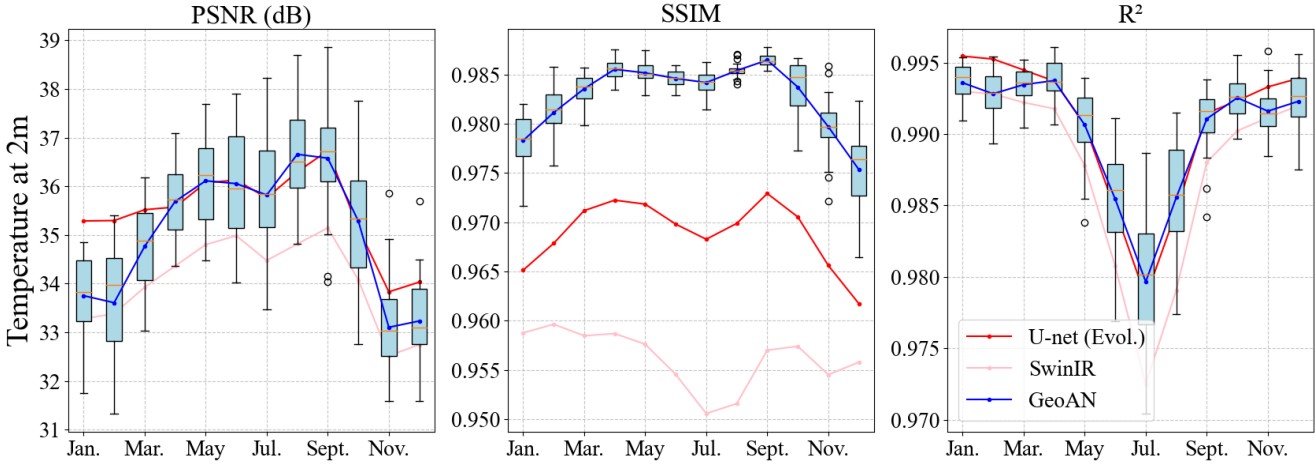

**Figure A1.** The monthly average statistic of GeoAN in 2023 compared with other methods on temperature (K) at 2m. The box plot is the distribution of GeoAN in each month.

## Appendix A: T2m comparison against U-net (Evol.)

Shown in Tab. 1, U-net (Evol.) outperformed GeoAN in T2m on PSNR and $R^2$. To understand these results, we analyzed the error distributions of T2m temporal and spatial in Fig. 6 and Fig. 7. During the summer season, the results of GeoAN have a similar performance with U-net (Evol.) in terms of PSNR and $R^2$ as shown in Fig. A1. The higher the altitude, the more error in GeoAN observed refers to Fig. 7. Furthermore, both high-altitude and winter conditions result in lower temperatures, leading us to conclude that GeoAN performs poorly in cold regions and during colder periods. To verify whether PSNR and $R^2$ react to the real performance in cold environments, we carried out an annual comparison in 2023 at high altitudes area, the specific results are shown in Fig. A2. In comparison, the texture of GeoAN is clearer than U-net (Evol.), and the temperature values in each pixel of these two methods are close, the difference is almost negligible. However, the improvement in sharpness GeoAN brings is discernible to the naked eye. As shown in Fig. 6, the largest RMSE between ground truth and GeoAN in winter is around 3K. The average RMSE of T2m is 1.40K, consider the RMSE between CLDAS and in-situ stations is 1.8K, the bias in GeoAN totally could be accepted.

## Appendix B: Discussion on how GeoAN restore the absent details

The PSNR of GeoAN on PRS achieves an impressive 47.251dB on the testing dataset. The leading performance compared with other methods of PRS is larger than other variables, and we find the PRS of GeoAN in mountainous areas outperforms others. GeoAN can produce the details that are not included in the input data as shown in Fig. 3. Considering the primal input data ERA5 lacks the details as shown in Fig. B1, it is reasonable to infer that deep learning methods can learn detailed information from the distribution of the ground truth in the training stage. The less range of change in the variables, the better the results

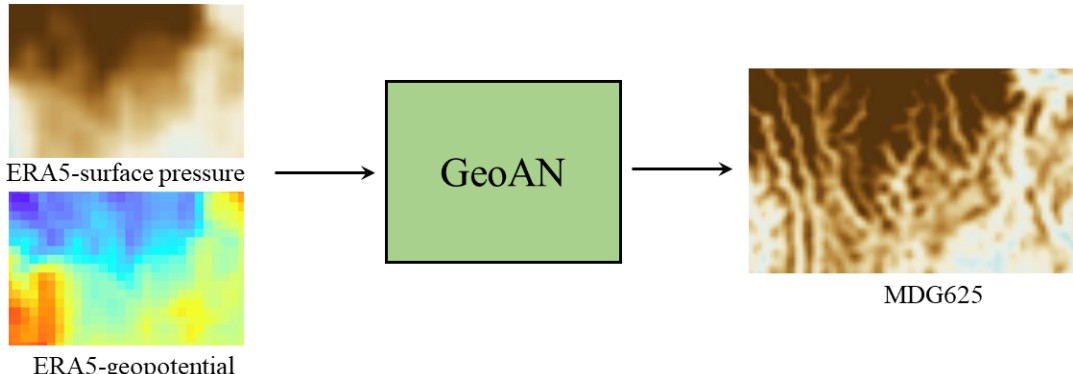

**Figure A2.** Temperature at 2m comparison between U-net (Evol.), GeoAN, and ground truth at high altitude areas (Himalayas areas) in 2023. Only the results of the first day of odd-numbered months are shown for convenient observation.

**Figure B1.** GeoAN can restore the details that are not included in the input data.

will be. During the training phase, adequate information about the spatial structure is furnished for GeoAN. Additionally, the low-resolution ERA5 input for each test case provides the value corresponding to each grid point. The ability of GeoAN to generate detailed features stems from these factors. Additionally, our statistics reveal that changes in pressure are significantly smaller compared to other variables, which likely explains why pressure metrics outperform others. The annual variation range in temperature surpasses that of pressure, therefore the PSNR value for T2m is inferior compared to PRS. Consequently, the poor performance of WS10m can be explained by its erratic fluctuations, as the training dataset lacks adequate information for models to accurately reconstruct its detailed characteristics.

*Author contributions.* SZJ and YLN designed the research. SZJ and CZX performed the experiments and code. SZJ wrote the manuscript. LYY, YSS, and ZXW process data analysis. LM and YLN provided the resource. LM and YLN supervised and reviewed the manuscript.

*Competing interests.* The contact author has declared that none of the authors has any competing interests.

*Disclaimer.* The MDG625 are produced by GeoAN, which is trained by ERA5 and CLDAS. To download and use the algorithms and
datasets associated with this paper, please follow the relevant restrictions and requirements. The license of GeoAN is Apache-2.0 and of MDG625 is CC-BY 4.0.

*Acknowledgements.* This document is the results of the research project funded by the International Research Center of Big Data for Sustainable Development Goals (No. CBAS2022GSP07), and the National Natural Science Foundation of China (No. 42230505, 42206148). We thank China Meteorological Administration and European Centre for Medium-Range Weather Forecasts for the provided datasets. We
thank all reviewers, editors, and others who helped with this paper.

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
