# Peer review of "MDG625: A daily high-resolution meteorological dataset derived by geopotential-guided attention network in Asia (1940-2023)"

_Earth System Science Data, 2024_

## Referee Comment (RC2)

The study downscaled a 6 km$^2$ daily meteorological dataset based on a geopotential-guided attention network (GeoAN) in Asia, covering the period from 1940 to 2023. The dataset is likely to be valuable for a wide range of applications, including climate change studies and extreme weather events analysis. However, the precision and validation of the downscaled dataset require more comprehensive evaluation. Additionally, the methodology could benefit from more detailed comparisons with alternative approaches to further highlight its advantages. The English presentation of the also requires significant improvement. Below are my specific comments:

1. The English presentation in the manuscript should be much more formal and precise. For example:
   - line 29: Replace 'kind of research' with a more formal term.
   - line 31: 'kinetic downscaling' should be corrected.
   - line 125: The phrase 'in this paper for the downscaling task, and SwinIR 125 can be found in' should be corrected.
   - Line 135: The phrase 'this phenomenon may caused by the that' contains a grammatical error, 'the' should be removed.
   - Line 76: why mentioned 'For the TP (mm), the day sum is adopted' again after line 70 ? consider removing line 76.

2. The introduction requires more comprehensive structured and detailed descriptions.
   - Lines 24–27 and 44-47: While these lines list several studies, the authors should provide more detailed information about each study and explain their relevance to this research.
   - The author should clearly outline the research gaps and demonstrate the unique advantages of their methodology.

3. For the methodology, the study only used daily data from 2020–2022 for training and daily data from 2023 for validation, to downscale 60 years of data since 1940. This raises concerns about whether the training and validation dataset is sufficient to ensure the accuracy of the downscaled dataset. I suggest:
   - Use a larger training and validation dataset to improve reliability.
   - Consider the leave-one-out cross-validation method to enlarge the training dataset.

4. Fo results,
   - Quantitative results (e.g., $R^2$, PSNR) demonstrate the effectiveness of GeoAN, but the discussion of why GeoAN outperforms other methods could be expanded.

- The visual comparisons (Fig.3-5) look impressive; however, it would be useful to provide more quantitative evidence to support claims about GeoAN's ability to restore fine details.

5. The decision not to include precipitation data is understandable but significant limits the dataset's utility for hydrological modelling applications. Future plans to address this gap would be valuable.

6. The Discussion section should give more information about:

   - Comparison of the methodology with other downscaling methods or downscaled datasets.
   - Advantages and limitations of the proposed methodology should be explicitly described.
   - The author could consider referencing recent methodologies, such as '*Multiple-point geostatistics-based spatial downscaling of heavy rainfall fields*' (Doi: 10.1016/j.jhydrol.2024.130899), to provide a broader context for their work.

---

## Author Comment (AC1)

Author Reply to RC1 of essd-2024-137

We are pleased to get the comment from the reviewer. Concerning the queries, here are our replies.

Q1: The English needs improvement, e.g., page 2, line 19 "due to the limited data density", what's the meaning of data density here? The terminology should be precise, e.g., page 2, line 31, kinetic downscaling should be dynamical downscaling.

A1:

(1) "Data density" here means the distributions of in-situ station, which mainly contains station observations. The distribution of stations cannot be unlimited dense, contributing to hardly producing a high-resolution dataset, especially in decades ago. Following is the revised description in manuscript: "…However, the distribution of in-situ stations is too sparse to produce a high-quality reanalysis dataset, especially for decades ago…"

(2) Thanks a lot for pointing out the usage of "dynamical downscaling", we have revised our manuscript and further polished the paper writing to avoid the same problems.

Q2: Page 2, line 32, "While the existing downscaling methods could produce high-resolution results, the results are unsatisfactory and unable to reconstruct detail and texture information." Please give the support and references for this statement about the limitations of existing methods.

A2:

As suggested by the reviewer, more details are introduced in the manuscript:

"…Dynamical downscaling methods are usually based on Regional Climate Models (RCMs) under the guidance of the initial fields produced by Global Climate Models (GCMs). Although the resolution of RCMs is higher than GCMs, the comprehensible ability to understand the real world is not enough. It leads to a considerable bias (Teutschbein and Seibert, 2012) due to the difficulty of establishing simulation equations that cannot meet the needs of various related tasks. From another opinion, the computational cost of RCMs is huge, and it is an obstacle to produce a wider range of results (Giorgi and Gutowski Jr, 2015; Di Luca et al., 2015). Compared with dynamical downscaling, statistical downscaling maps the relationship between high-resolution and low-resolution from historical data to produce results. The computational cost and bias of statistical downscaling are lower than dynamical downscaling methods…Using deep learning methods to downscale the geographic data can effectively avoid the problems encountered by former downscaling methods, such as high biases, regional sensitivity, high computational cost, etc. Deep learning methods use deep layers to bridge the relationship between low-resolution and high-resolution data. As a result, robustness against the sensitivity can be achieved with the increasing amount of training data. Once the model has been trained, the computational cost is at a low level during the using step, and the deep learning method can nest a wide range easily…"

Q3: The literature review in the introduction should be more comprehensive. The current state of research has not been adequately presented. The authors are suggested to highlight the limitations of existing methods rather than merely listing several studies. For example, the authors list several works using the Transformer architecture without presenting their relationship to the work in this paper.

A3:

(1) Following the suggestions from the reviewer, we presented more details of the literature review, the revised version is as follows:

"…On the one hand, Liang et al. (2021) proposed SwinIR and achieved impressive results in the SR task which can be considered as the benchmark for the SR (Super-Resolution) task. The core algorithm of SwinIR is to use no overlap windows in order to split the input feature for calculating the attention relationship inner each window, then shift the windows by the step of half-width of the windows and calculate the relationship again. Meanwhile, Song and Zhong (2022) proposed a novel network to harvest long-range information from global instead of inner the window. The experimental results on SR benchmarks (Bevilacqua et al., 2012; Martin et al., 2001; Huang et al., 2015; Matsui et al., 2017) show this strategy can achieve better results… Shen et al. (2023) proposed a near-surface air temperature downscaling network SNCA-CLDASSD. In this model, Shen et al. used two attention blocks to downscale the input data called Cross-Attention based on Light-CLDASSD. However, only near-surface air temperature (temperature at 2 meters) was considered in this work and the network was built on CLDAS, which cannot cover long-term years. On the other hand, Liu et al. (2023) used the terrain to guide the deep learning network for the downscaling task called terrain-guided attention network (TGAN) in Southwest China. TGAN used the digital elevation model (DEM) to build high-resolution temperature (temperature at 2 meters, the same as SNCA-CLDASSD) results. The data range of TGAN used began in 2018 and TGAN also cannot be used in the historical situation. What's more, Zhong et al. (2023) proposed a transformer-based learning method Uformer, which used topography data to achieve high-resolution meteorological variables in inner Mongolia province, China. Although topography data can help rebuild the high-resolution, adding into the input low-resolution directly will lose the characters of topography. All of the above, existing advanced deep learning methods of meteorological downscaling mostly employ attention architecture (Transformer is one of the special attention architectures) …"

(2) Transformer architecture has been widely used and considered as the most advanced unit in deep learning methods in recent years due to its excellent performance. Existing advanced deep learning methods of meteorological downscaling mostly used attention architecture (Shen et al., 2023; Liu et al., 2023; Zhong et al., 2023;). However, now existing deep learning downscaling methods only focus on one or two meteorological variables, while different variables have correlations and deep learning could handle multiple variables simultaneously. Lastly, it's worth noticing that there are no models that can cover a long-term and wide range of historical multiple variables. Our work is under the basis of the transformer architecture, and we merge geopotential into the transformer block to enhance the geographic information for multiple meteorological variables not only to save the computational resources but also to improve the performance of the model. And we produce a long-term historical dataset to fill the gap in high-resolution historical meteorological datasets.

Q4: The authors are suggested to present the motivations of using the variables of temperature at 2m, pressure at the surface, and wind speed at 10m for GeoAN in this work.
A4:
Nowadays, existing methods usually utilize W10m and T2m as the downscaling variables (Shen et al., 2023; Liu et al., 2023; Zhong et al., 2023;). We compared the variables between CLDAS and ERA5 land surface dataset, and chose the shared variables in both datasets including temperature at 2m, pressure at the surface, wind speed at 10m, and precipitation. Then we evaluated the precipitation of CLDAS and ERA5. Due to the low correlation of precipitation in these two datasets, we discarded the precipitation and used the rest three variables. Noted that the wind speed provided in ERA5 is divided into two components of u and v at 10m and in CLDAS is the synthesized wind speed. We calculated the

synthesized wind speed using the data provided by ERA5 and perform the following steps.

Q5: Page 6, line 98, "GeoAB is repeated 18 times", What considerations are there for setting it to 18?
A5:
On the one hand, deep learning methods commonly use repeated blocks to reach a deep network for harvesting deep information, and that's why it is called "deep learning". In this theory, the deeper the network is, the more performance it will have. However, when the network is too deep, the chain rule will cause exploding or vanishing gradient problems. On the other hand, the computational expense of deep networks is huge and the training step will last several weeks. We lastly choose 18 as our repeating times, and it serves as the result of balancing the computational resources and the depth of the network. If we choose a deeper network, our GPU (4 RTX 6000 Ada GPU 48G) can't afford the computational expense and may cause exploding or vanishing gradient problems.

Q6: Page 6, line 105, "GUP memory limitation" should be "GPU memory limitation"?
A6:
Yes, here is a typo, it should be "GPU memory limitation". We will revise this mistake in the next version and check the whole paper to avoid the same mistake.

Q7: In addition to UNet and SwinIR, the authors are suggested to compared the results of the proposed method with more existing representative deep learning based downscaling methods.
A7:
As far as we are awarded, our work is the first attempt to establish the mapping relationship from ERA5 to CLDAS to make a long-term meteorological dataset. Thus, there are no existing works that can be compared directly. Downscaling is a similar task to super-resolution in computer vision, so we choose three super-resolution methods migrating to this task to make a comparison:
1) Bilinear is the most classic and commonly used algorithm to improve resolution.
2) U-net has been one of the most used deep learning methods in recent years in various tasks for its strong universality.
3) SwinIR, which was proposed in 2021, was a novel and SOTA (state of the art) super-resolution model based on transformer block, which can be considered as the benchmark as well as the representative method of super-resolution missions.

In order to migrate the methods to downscale the long-term meteorological dataset from ERA5 to CLDAS, we have to redesign each existing method. These methods may take several weeks to months to retrain so as to be compatible with the task.
In the future, we will continue to make larger comparisons with more existing representative deep learning methods from different fields and tasks, and delve into how these deep learning methods can dig geographic information to help various geographic tasks.

Q8: The proposed method has been used to generate a long-term dataset; however, the validation dataset in this paper is quite limited. It is suggested to evaluate the accuracy and reconstruction quality in terms of PSNR and SSIM using a larger dataset.
A8:
We have downloaded the CLDAS dataset from 2020 to 2023. The validation dataset needs to cover one

full cycle, for meteorological task is one year. The amount of training data is directly related to the performance of the network. Under guaranteeing data spanning at least a whole year for validation, we used as much as we can to train the model. Thus, we used the entire data from 2023 to validate the proposed method and used data from 2020 to 2022 to train our own work. Although CLDAS contains the data since 2008, it's hard to obtain previous data after 2017, thus we cannot access enough long-range data to validate our model. What's more, the increase in the amount of data can also greatly expand and prolong the network's training expenditure and time. In future work, we will try to use other wider range of datasets to validate geography-related tasks.

**References:**

Bevilacqua, M., Roumy, A., Guillemot, C., and Alberi-Morel, M. L.: Low-complexity single-image super-resolution based on nonnegative neighbor embedding, in: The British Machine Vision Conference, pp. 1–10, 2012.

Di Luca, A., de Elía, R., and Laprise, R.: Challenges in the quest for added value of regional climate dynamical downscaling, Current Climate Change Reports, 1, 10–21, 2015.

Giorgi, F. and Gutowski Jr, W. J.: Regional dynamical downscaling and the CORDEX initiative, Annual review of environment and resources, 40, 467–490, 2015.

Huang, J.-B., Singh, A., and Ahuja, N.: Single image super-resolution from transformed self-exemplars, in: Proceedings of the IEEE Conference on Computer Vision and Pattern Recognition, pp. 5197–5206, 2015.

Liang, J., Cao, J., Sun, G., Zhang, K., Van Gool, L., and Timofte, R.: Swinir: Image restoration using swin transformer, in: Proceedings of the IEEE/CVF International Conference on Computer Vision, pp. 1833–1844, 2021.

Liu, G., Zhang, R., Hang, R., Ge, L., Shi, C., and Liu, Q.: Statistical downscaling of temperature distributions in southwest China by using terrain-guided attention network, IEEE Journal of Selected Topics in Applied Earth Observations and Remote Sensing, 16, 1678–1690, 2023.

Martin, D., Fowlkes, C., Tal, D., and Malik, J.: A database of human segmented natural images and its application to evaluating segmentation algorithms and measuring ecological statistics, in: Proceedings of the IEEE/CVF International Conference on Computer Vision, vol. 2, pp. 416–423, IEEE, 2001.

Matsui, Y., Ito, K., Aramaki, Y., Fujimoto, A., Ogawa, T., Yamasaki, T., and Aizawa, K.: Sketch-based manga retrieval using manga109 dataset, Multimedia Tools and Applications, 76, 21 811–21 838, 2017.

Song, Z. and Zhong, B.: A Lightweight Local-Global Attention Network for Single Image Super-Resolution, in: Proceedings of the Asian Conference on Computer Vision, pp. 4395–4410, 2022.

Shen, Z., Shi, C., Shen, R., Tie, R., and Ge, L.: Spatial Downscaling of Near-Surface Air Temperature Based on Deep Learning Cross-Attention Mechanism, Remote Sensing, 15, 5084, 2023.

Teutschbein, C. and Seibert, J.: Bias correction of regional climate model simulations for hydrological climate-change impact studies: Review and evaluation of different methods, Journal of hydrology, 456, 12–29, 2012.

Zhong, X., Du, F., Chen, L., Wang, Z., and Li, H.: Investigating transformer-based models for spatial downscaling and correcting biases of near-surface temperature and wind-speed forecasts, Quarterly Journal of the Royal Meteorological Society, 2023.

---

## Author Comment (AC2)

**Author Reply to RC2 of essd-2024-137**

We sincerely appreciate the time and effort Referee #2 has dedicated to reviewing our work and providing valuable feedback. It is a great honor to receive such insightful comments, which have helped us improve and refine our manuscript. We are truly grateful for your patience and thoughtful guidance. May our replies answer the questions raised in your comment. Below, we offer detailed responses to the queries and suggestions raised.

**General comments:**

The study downscaled a 6 km$^2$ daily meteorological dataset based on a geopotential-guided attention network (GeoAN) in Asia, covering the period from 1940 to 2023. The dataset is likely to be valuable for a wide range of applications, including climate change studies and extreme weather events analysis. However, the precision and validation of the downscaled dataset require more comprehensive evaluation. Additionally, the methodology could benefit from more detailed comparisons with alternative approaches to further highlight its advantages. The English presentation of the also requires significant improvement.

**General responses:**

We appreciate for reviewing our work and giving your valuable suggestions. We mainly compared our results with CLDAS, a high-quality land assimilation dataset in Asia. Since it's hard to obtain grid true value to compare in a long historical term, we will seek more cooperation for validation and using a large range of data, including remote sensing, gauged value, and so on, in the future. As to the reasons for outperformance, we try to explain more comprehensively why GeoAN performs better by analyzing each method's visual results and how the deep learning methods restore the fine details from the coarse input. The model could learn geographic world information by using geopotential from the distribution of the training dataset. For more information can refer to **A4** below. Lastly, we carefully revised the manuscript and polished the English presentation following the native speakers' suggestions.

**Q1:** The English presentation in the manuscript should be much more formal and precise.
- Line 29: Replace 'kind of research' with a more formal term.
- Line 31: 'kinetic downscaling' should be corrected.
- Line 125: The phrase 'in this paper for the downscaling task, and SwinIR can be found in' should be corrected.
- Line 135: The phrase 'this phenomenon may caused by the that' contains a grammatical error, 'the' should be removed.
- Line 76: why mentioned 'For the TP (mm), the day sum is adopted' again after line 70 ? consider removing line 76.

**A1:** Thank you so much for pointing out our inappropriate paper writing, we will carefully address the grammatical issues and further refine the manuscript. Below are our detailed responses to the points mentioned.

**Point 1:** We used *"various studies"* to replace the original *"kinds of research"*:

*Lines 32 in the revised version:*

*"High-quality and high-resolution data is necessary for various studies, to solve the contradiction."*

**Point 2:** The incorrect usage of *"kinetic downscaling"* in the whole manuscript has been corrected to *"dynamical downscaling"* in **lines 36, 41, and 43** of the revised manuscript.

**Point 3:** The Original phrase was refined as follows:

*Lines 148-156 in the revised version:*

*"This section details experiments to evaluate the performance of GeoAN. For comparison, the classic algorithm bilinear interpolation, widely used in downscaling, is included. Additionally, two deep learning methods, U-Net (Ronneberger et al., 2015) and SwinIR (Liang et al., 2021), were employed for comparative analysis. The source code for both networks was obtained from their perspective GitHub repositories. To ensure a fair comparison, the U-Net architecture was modified for the downscaling task, resulting in a customized version referred to as U-Net Evolution (U-Net Evol.). The original U-Net implementation is available at https://github.com/milesial/Pytorch-UNet, while the SwinIR code can be accessed at https://github.com/JingyunLiang/SwinIR. To maintain consistency, all deep learning models were configured with equivalent parameters or computational complexity, and they were trained for 100 epochs using identical hyperparameters under the same environmental conditions."*

**Point 4:** The redundant *"the"* has been removed from the phrase:

*Lines 163 in the revised version:*

*"...this phenomenon may caused by that..."*

**Point 5:** We mentioned the values of PRS, T2m, and WS10m are calculated from the mean results. Corresponding to the above information, we explain that TP is the total precipitation sum up in one whole day again. It seems that the content is indeed repeated and we refine the text as follows:

*Lines 101-103 in the revised version:*

*"The temporal resolution of these two datasets is calculated to one day, which is calculated by the mean of PRS (hPa), T2m (K), WS10m (m2/s) and the sum of TP (mm) over the whole day respectively using the original hourly data."*

**Q2:** The introduction requires more comprehensive structured and detailed descriptions.

- Lines 24–27 and 44-47: While these lines list several studies, the authors should provide more detailed information about each study and explain their relevance to this research.

- The author should clearly outline the research gaps and demonstrate the unique advantages of their methodology.

**A2:** We sincerely appreciate your constructive and insightful suggestions for improving the manuscript. We have added an introduction to the related work part in the paper and provided a detailed discussion on the development of downscaling. Below, we outline our revisions along with the corresponding new line numbers for your reference regarding the work suggested.

**Point 1:**

*Lines 23-30 in the revised version:*

*"He et al. (2020) produced a meteorological dataset with a spatial resolution of 0.1° from 1979 in China. In this paper, the China Meteorological Forcing Dataset was proposed by fusing remote sensing products, reanalysis datasets, and in-situ station data. The most significant contribution of this work was using a larger number of stations to raise the quality of the dataset. A long-term gridded daily meteorological dataset for northwestern North America was proposed by Werner et al. (2019). The authors try to produce a dataset for training statistical downscaling schemes in Canada. The same in Italian, Bonanno et al. (2019) proposed the high-resolution meteorological*

*dataset named MERIDA. MERIDA was produced by dynamical downscaling from the fifth-generation reanalysis dataset for the global climate and weather (ERA5) using WRF."*

*"Shen et al. (2023) proposed a near-surface air temperature downscaling network SNCA-CLDASSD. In this model, Shen, et al. used two attention blocks to downscale the input data called Cross-Attention based on Light-CLDASSD. However, only near-surface air temperature is considered in this work and the network was built on CLDAS, which cannot cover long-term years. Liu et al. (2023) used the terrain to guide the deep learning network for the downscaling task called terrain-guided attention network (TGAN) in Southwest China. TGAN used the digital elevation model (DEM) to build high-resolution temperature (at 2 meters) results. The range of TGAN used begins in 2018 and TGAN cannot be used in the historical situation. Zhong et al. (2023) proposed a transformer-based learning method Uformer, which directly adds topography data, to achieve high-resolution meteorological variables in inner Mongolia province, China. Although topography data can help rebuild the high-resolution, directly adding into the input low-resolution will lose the characters of topography. All of the above, existing advanced deep learning methods of meteorological downscaling mostly used attention architecture (Transformer is one of the special attention architectures). However, now existing methods focus on one or two meteorological variables, while different variables have correlations and deep learning could handle multiple variables simultaneously. This way, not only saves the computational resources but also improves the performance of the model. Last and most important, there are no models that can cover a long-term and wide range of historical multiple variables."*

**Point 2:** We appreciate the insightful review and it truly helps us improve the expression and can explain the research characteristics clearly. The appended discussion above for providing more outline information is added in the introduction section of the revised manuscript shown as follows:

*"In this paper, we propose a new attention-based network called the Geopotential-guided Attention Network (GeoAN), the structure of which is shown in Fig. 1 for downscaling meteorological variables, including temperature at 2m (T2m), pressure at the surface (PRS), and wind speed at 10m (WS10m) from 0.25° to 0.0625°. The proposed GeoAN is guided by the geopotential, which makes the model learn information directly instead of spontaneously… The data quality and resolution of CLDAS are relatively high, but data is only available for China and the surrounding areas and for the years after 2008. After utilizing deep learning networks to construct the mapping relationship between ERA5 and CLDAS, a historical meteorological dataset since 1940 was produced. Our produced MDG625 makes up for the lack of the CLDAS before 2008 and increases the spatial resolution of ERA5."*

**Q3:** For the methodology, the study only used daily data from 2020–2022 for training and daily data from 2023 for validation, to downscale 60 years of data since 1940. This raises concerns about whether the training and validation dataset is sufficient to ensure the accuracy of the downscaled dataset. I suggest:
- Use a larger training and validation dataset to improve reliability.
- Consider the leave-one-out cross-validation method to enlarge the training dataset.

**A3:** Thank you so much for the insightful suggestions. Using more data is absolutely a good way to increase the performance. In the future, we will try to use a wider year range for training models. However, here are some considerations about how many years we should use suitable for this work we have consideration.

**Point 1:**

**1)** The data is used to train the model for building the relationship between CLDAS and ERA5, when the training step is finished, the relation between low and high resolution is fixed. The performance is only affected by how well the model learns, no matter how many years MDG625 contains. Using more data needs a larger model to adapt, and the small model makes it hard to learn so much information from a huge dataset. However, training a large model is a big challenge for both hardware and time costs. Another reason we use these years is also considered, the huge performance raised by the increasing data size usually needs the increasing order of magnitude of the data, and for a slight performance increase, a huge cost is not efficient. We will produce MDG625 V2 with a stronger and larger model using a larger dataset in the future, including reanalysis datasets, remote sensing, and other data sources.

**Table AC2-1.** The data range of different deep learning downscaling methods is used.

|  | Training dataset | Testing dataset |
| --- | --- | --- |
| Zhong et al. (2023) | 2020.01 – 2021.05 | 2021.06 – 2021.09 |
| Liu et al. (2023) | 2019.01 – 2019.12 | 2018 |
| Shen et al. (2023) | 90% of 2016, 2017, 2019, 2020 | 2018 |

**2)** Due to many researchers (Zhong et al. (2023); Liu et al. (2023); Shen et al. (2023)), as shown in Tab. AC2-1, only using the data in one or two years for downscaling one or two meteorological parameters, we finally decided to use 3 years for training. The total size stored in 'numpy format (.npy)' is up to around 40G (30G for training+10G for testing). Considering the size of the high-resolution data (1040×1600) is big when comparing other deep learning methods, the number of training datasets is reasonable.

**3)** The validation dataset needs to cover one full cycle, for the meteorological task is one year. Although there are slight distributing differences between different years, using a whole year of data for validation is enough to cover most situations.

**Point 2:** The cross-validation method is known for its efficient use of data and reducing overfitting. When addressing the one-shot task, it is a powerful technique to adopt. However, the cross-validation method is not a common method in deep learning since there are other skills to avoiding overfitting like regularization or dropout. We followed the most experiment designs and skills in the super-resolution (SR) and that is why there is no cross-validation in this work. On the one hand, there are enough data and training epochs to learn the relationship. On the other hand, the progress of training a model is complex, and for a better result, most researchers usually use lots of skills and pre-trained models to accelerate model convergence, and it is hard to make progress in cross-validation in deep learning work. We used the pre-trained models from the former experiments too as shown in Fig. AC2-1. As shown in the figure, our proposed model's convergence is satisfactory after 100 training epochs and learns well from the dataset.

[Figure]

**Figure AC2-1.** The original training logs of GeoAN. The final GeoAN is trained from the pre-trained model: geoan-2024-0412-1935-4239 (recorded as version-4239), and the version-4239 is trained from version-4399.

**Q4:** For results,

- Quantitative results (e.g., R², PSNR) demonstrate the effectiveness of GeoAN, but the discussion of why GeoAN outperforms other methods could be expanded.
- The visual comparisons (Fig.3-5) look impressive; however, it would be useful to provide more quantitative evidence to support claims about GeoAN's ability to restore fine details.

**A4:** Thanks a lot for the suggestions. Deep learning is criticized for its unexplained ability, but also achieves great results (Chakraborty et al., 2017). It's a big challenge for us to explain why GeoAN, a deep learning method, performs at a mechanistic level. However, we still try to explain how GeoAN outperforms others.

**Point 1: Discussion on why GeoAN performs better than other methods**

It is hard to say which skill contributes most to the performance because the explanation is lacking in deep learning. Analyzing the experiment's progress, the biggest difference of GeoAN is the geopotential-guided attention block (GeoAB) we proposed in this work. The geopotential can reflect many geographic information and it helps the model understand the world well. In other words, the model can know such as elevation directly without learning. Elevation is closely related to the meteorological variables definitely and GeoAB can guide the model to learn the information associated with the variables directly. In other compared methods, there is no geopotential to guide producing progress, and these methods need to learn the elevation relationship spontaneously. It's more like an engineering problem to get the best result and produce the historical dataset.

[Figure]

**Figure AC2-2.** GeoAN can restore the details that are not included in the input data.

**Point 2: Discussion on why PSR is superior to other variables and how GeoAN restores fine details**

We discuss this part in Appendix B in the revised version, and here is the content of the discussion:

*Lines 251-261 in the revised version:*

*"The PSNR of GeoAN on PRS is up to 47.251dB on the testing dataset. The leading performance compared with other methods of PSR is larger than other variables, and we find the PRS's outperformance of GeoAN in mountainous areas extremely obvious. GeoAN can produce the details that are not included in the input data as shown in Fig. 3. Considering the original input data ERA5 lacks the details as shown in Fig. AC2-2, it is reasonable to infer that deep learning methods can learn detailed information from the distribution of the ground truth in the training step. The less change in the variables, the better the results will be. The training step provides enough information on the detailed shape and the low-resolution input ERA5 of each test offers the value of each grid. This is the most important reason why GeoAN can produce fine details. We also found the change in pressure is much lower than other variables, and we infer that's why the metrics of pressure perform better than others. The change of temperature in one year is bigger than the pressure as shown in Fig. AC2-3 and AC2-4, and the PSNR of T2m is worse than PRS. Therefore, the performance of WS10m is worst can be inferred from the drastic irregular changes, because there is not enough information for the models to restore the details of WS10m in the training dataset."*

[Figure]

**Figure AC2-3.** The PRS details of GeoAN (above one) and Ground Truth (below one). There are 6 days listed in the figure distributed in 2023. The complete information on the area can be found in the paper's Fig. 3. The change in PRS is very small in one year, and each result of these 6 days is similar.

[Figure]

**Figure AC2-4.** The T2m details of GeoAN (above one) and Ground Truth (below one) in the same area of PRS in Fig. AC2-3. Compared with the change in one year of PRS, the change in T2m is larger.

**Q5:** The decision not to include precipitation data is understandable but significantly limits the dataset's utility for hydrological modeling applications. Future plans to address this gap would be valuable.

**A5:** We agree with this suggestion very much, we are concerned about the precipitation since its importance. Here are some future plans we considered in future.

**1)** We are going to try more sources of data for precipitation not only the reanalysis production, but especially remote sensing, radar, and gauged stations.

**2)** The merging geographic mechanism in the deep learning model is the most promising solution to downscale higher-resolution precipitation. In the future, this will be our main research direction.

**3)** Precipitation is not closely related to geopotential; we will further find other geographic elements to guide the deep learning method to downscale the precipitation.

**4)** The forecasting of precipitation is valuable, and we will try to produce a forecasting system that not only produces the historical dataset.

The above discussion can be found in the revised manuscript:

*Lines 230-232 in the revised version:*

*"In future work, we will try to merge remote sensing and gauged precipitation to downscale the precipitation using other geographic principles and using more suitable variables instead of geopotential."*

**Q6:** The Discussion section should give more information about:
- Comparison of the methodology with other downscaling methods or downscaled datasets.

The advantages and limitations of the proposed methodology should be explicitly described.

- The author could consider referencing recent methodologies, such as 'Multiplepoint geostatistics-based spatial downscaling of heavy rainfall fields' (Doi:10.1016/j.jhydrol.2024.130899), to provide a broader context for their work.

**A6:** Your suggestion is insightful for us to improve this work. As to the questions in the discussion, here are our replies:

**Point 1:** As far as we are known, our work is the first attempt to establish the mapping relationship from ERA5 to CLDAS to make a long-term daily meteorological dataset. Thus, there are no existing works that can be compared directly. Downscaling is a similar task to super-resolution in computer vision, so we choose three super-resolution methods migrating to this task to make a comparison:

1) Bilinear is the most classic and commonly used algorithm to improve resolution.

2) U-net has been one of the most used deep learning methods in recent years in various tasks for its strong universality.

3) SwinIR, proposed in 2021, was a novel SOTA (state-of-the-art) super-resolution model based on a transformer block. It can be considered the benchmark and representative deep learning method of SR-related missions.

To migrate the methods to downscale the long-term meteorological dataset from ERA5 to CLDAS, each existing method has to be redesigned. These methods may take several weeks to months to retrain to be compatible with the task. In the future, we will continue to make larger comparisons with more existing representative deep learning methods and delve into how these deep learning methods can dig geographic information to help downscale the variables.

**Point2:** The advantages and limitations have been appended in the paper:

*Lines 210-216 in the revised version:*

*"However, GeoAN achieved a great result and MDG625 is valuable for a wide range of applications, there are some limitations in this work. Limited by the computational expenses and the memory of GPU, we cannot use a larger model to train with more datasets. A larger model and more training datasets could help the performance a lot. The MDG625 produced the area in Asia, however, if the fixed position information was masked and more datasets to cover the other areas in the training step, the global results could be produced. A global high-resolution historical dataset can be very valuable. In addition, by our proposed GeoAN, hourly variables also can be produced, but only daily data are provided in MDG625 since the hard disk capacity is not enough for such a large amount of data."*

*Lines 223-230 in the revised version:*

*"Considering the rarity of long-term historical high-resolution meteorological data in Asia, MDG625 (Meteorological Dataset with 0.0625° resolution produced by a Geopotential-guide attention network) provided a solution using a deep geographic coupling attention network called the geopotential-guide attention network (GeoAN) within an acceptable error. GeoAN could learn the geopotential relationship directly, which is closely related to meteorological variables. This Strategy could help the network understand the geographic world more easily and produce a better result. MDG625 contains daily temperature at 2m, surface pressure, and 10m wind speed since 1940. Experimental results demonstrated the superior performance of the GeoAN and the satisfaction of MDG625. Our proposed MDG625 could make up for the*

*lack of a long historical meteorological high-resolution dataset. For various meteorological researches is valuable."*

**Point3:** After reading the recommended paper, it contains more discussion on precipitation downscaling.

This part will be discussed and the recent work will be referred to in the revised manuscript:

*Lines 216-221 in the revised version:*

*"Lastly, precipitation poses a challenge for statistical models, particularly in extreme cases (Zou et al., 2024; Sachindra et al., 2018; Xu et al., 2015). The dataset referenced in MDG625 includes three meteorological variables (PRS, T2M, and WS10m), but it lacks precipitation data. The low correlation of total precipitation (TP) between the ERA5 and CLDAS datasets is the primary reason why the GeoAN model struggles to accurately restore the spatial structure of precipitation. Inspired by the work of (Zou et al., 2024), more comprehensible algorithms could be implemented in GeoAN in the future."*

**References:**

Bonanno, R., Lacavalla, M., and Sperati, S.: A new high-resolution meteorological reanalysis Italian dataset: MERIDA, Quarterly Journal of the Royal Meteorological Society, 145, 1756–1779, 2019.

Chakraborty, S., Tomsett, R., Raghavendra, R., Harborne, D., Alzantot, M., Cerutti, F., Srivastava, M., Preece, A., Julier, S., Rao, R. M., et al.:260 Interpretability of deep learning models: A survey of results, in: 2017 IEEE smartworld, ubiquitous intelligence & computing, advanced & trusted computed, scalable computing & communications, cloud & big data computing, Internet of people and smart city innovation (smartworld/SCALCOM/UIC/ATC/CBDcom/IOP/SCI), pp. 1–6, IEEE, 2017.

He, J., Yang, K., Tang, W., Lu, H., Qin, J., Chen, Y., and Li, X.: The first high-resolution meteorological forcing dataset for land process studies over China, Scientific data, 7, 25, 2020.

Liu, G., Zhang, R., Hang, R., Ge, L., Shi, C., and Liu, Q.: Statistical downscaling of temperature distributions in southwest China by using terrain-guided attention network, IEEE Journal of Selected Topics in Applied Earth Observations and Remote Sensing, 16, 1678–1690, 2023.

Sachindra, D., Ahmed, K., Rashid, M. M., Shahid, S., and Perera, B.: Statistical downscaling of precipitation using machine learning techniques, Atmospheric research, 212, 240–258, 2018.

Shen, Z., Shi, C., Shen, R., Tie, R., and Ge, L.: Spatial Downscaling of Near-Surface Air Temperature Based on Deep Learning Cross-Attention Mechanism, Remote Sensing, 15, 5084, 2023.

Shi, C., Xie, Z., Qian, H., Liang, M., and Yang, X.: China land soil moisture EnKF data assimilation based on satellite remote sensing data, Science China Earth Sciences, 54, 1430–1440, 2011.

Shi, C., Jiang, L., Zhang, T., Xu, B., and Han, S.: Status and plans of CMA land data assimilation system (CLDAS) project, in: EGU General Assembly Conference Abstracts, p. 5671, 2014.

Sun, S., Shi, C., Pan, Y., Bai, L., Xu, B., Zhang, T., Han, S., and Jiang, L.: Applicability assessment of the 1998–2018 CLDAS multi-source precipitation fusion dataset over China, Journal of Meteorological Research, 34, 879–892, 2020.

Werner, A., Schnorbus, M., Shrestha, R., Cannon, A., Zwiers, F., Dayon, G., and Anslow, F.: A long-term, temporally consistent, gridded daily meteorological dataset for northwestern North America, Scientific Data, 6, 1–16, 2019.

Xu, S., Wu, C., Wang, L., Gonsamo, A., Shen, Y., and Niu, Z.: A new satellite-based monthly precipitation downscaling algorithm with non-stationary relationship between precipitation and land surface characteristics, Remote sensing of environment, 162, 119–140, 2015.

Zhong, X., Du, F., Chen, L., Wang, Z., and Li, H.: Investigating transformer-based models for spatial downscaling and correcting biases of near-surface temperature and wind-speed forecasts, Quarterly Journal of the Royal Meteorological Society, 2023.

---

## Author Comment (AC3)

**Author Reply to RC1 of essd-2024-137**

We are truly grateful to Referee #1 for reviewing our manuscript and offering constructive feedback. These insightful comments played a key role in improving the quality of our work. We sincerely appreciate your patience and valuable thinking. As suggested by the comments, we revised our manuscript and we hope the following responses may adequately address the questions raised in the review.

**Q1:** The English needs improvement, e.g., page 2, line 19 "due to the limited data density", what's the meaning of data density here? The terminology should be precise, e.g., page 2, line 31, kinetic downscaling should be dynamical downscaling.

**A1:** "Data density" here means the distributions of in-situ stations (such as gauge), which mainly contain observation value. The distribution of stations cannot be unlimited dense, contributing to hardly producing a high-resolution dataset, especially before the 2000s. Following is the revised description in the manuscript:

> *Lines 36-44 in the revised version:*
>
> *"However, the distribution of in-situ stations is too sparse to produce a high-quality reanalysis dataset, especially for decades ago."*

Thanks a lot for pointing out the usage of *"dynamical downscaling"*, we have revised our manuscript and further polished the paper writing to avoid the same problems. The incorrect usage of *"kinetic downscaling"* in the whole manuscript has been corrected to *"dynamical downscaling"* in **lines 36, 41, and 43** of the revised manuscript.

**Q2:** Page 2, line 32, "While the existing downscaling methods could produce high-resolution results, the results are unsatisfactory and unable to reconstruct detail and texture information." Please give the support and references for this statement about the limitations of existing methods.

**A2:** Thanks for the insightful suggestions, as suggested by the reviewer, more details about the related works are introduced in the manuscript:

> *Lines 36-44 in the revised version:*
>
> *"Dynamical downscaling methods are usually based on Regional Climate Models (RCMs) under the guidance of the initial fields produced by Global Climate Models (GCMs). Although the resolution of RCMs is higher than GCMs, the comprehensible ability to understand the real world is not enough. It leads to a considerable bias (Teutschbein and Seibert, 2012) due to the difficulty of establishing simulation equations that cannot meet the needs of various related tasks. From another opinion, the computational cost of RCMs is huge, and it is an obstacle to producing a wider range of results (Giorgi and Gutowski Jr, 2015; Di Luca et al., 2015). Compared with dynamical downscaling, statistical downscaling maps the relationship between high-resolution and low-resolution from historical data to produce results. The computational cost and bias of statistical downscaling are lower than dynamical downscaling methods."*
>
> *Lines 53-57 in the revised version:*
>
> *"Super-resolution tasks are similar to geographic downscaling tasks, using SR deep learning*

*methods to downscale the geographic data can effectively avoid the problems encountered by former downscaling methods, such as high biases, regional sensitivity, high computational cost, etc. Deep learning methods use deep layers to bridge the relationship from low-resolution to high-resolution data and have robustness against the sensitivity. The computational cost is very low once the model has been trained, during the using step, and the deep learning method can nest a wide range easily."*

**Q3:** The literature review in the introduction should be more comprehensive. The current state of research has not been adequately presented. The authors are suggested to highlight the limitations of existing methods rather than merely listing several studies. For example, the authors list several works using the Transformer architecture without presenting their relationship to the work in this paper.

**A3:** The valuable comments are meaningful to improve our work, we will enhance the article following the suggestions. Existing advanced deep learning methods of meteorological downscaling mostly used attention architecture (Shen et al., 2023; Liu et al., 2023; Zhong et al., 2023;). However, now existing deep learning downscaling methods only focus on one or two meteorological variables, while different variables have correlations and deep learning could handle multiple variables simultaneously. After that, it's worth noticing that there are no models that can cover a long-term and wide range of historical multiple variables due to the limitation of the training dataset. Our work merged geopotential into the transformer block to enhance the geographic information for multiple meteorological variables not only saving the computational resources but also improving the performance of the model. And we produce a long-term historical dataset to fill the gap in high-resolution historical meteorological datasets. For more introduction of the related work and the limitations, we revised the manuscripts following the suggestions from the reviewer, the revised version is as follows:

*Lines 23-30 in the revised version:*

*"He et al. (2020) produced a meteorological dataset with a spatial resolution of 0.1° from 1979 in China. In this paper, the China Meteorological Forcing Dataset was proposed by fusing remote sensing products, reanalysis datasets, and in-situ station data. The most significant contribution of this work was using a larger number of stations to raise the quality of the dataset. A long-term gridded daily meteorological dataset for northwestern North America was proposed by Werner et al. (2019). The authors try to produce a dataset for training statistical downscaling schemes in Canada. The same in Italian, Bonanno et al. (2019) proposed the high-resolution meteorological dataset named MERIDA. MERIDA was produced by dynamical downscaling from the fifth-generation reanalysis dataset for the global climate and weather (ERA5) using WRF."*

*Lines 48-71 in the revised version:*

*"Liang et al. (2021) proposed SwinIR and achieved impressive results in the SR task and be considered the benchmark for the SR task. The core algorithm of SwinIR uses no overlap windows to split the input feature to calculate the attention relationship inner each window and shift the windows by the step of the half-width of the windows. Song and Zhong (2022) proposed a novel network to harvest long-range information from global instead of inner the window. The experimental results on SR benchmarks (Bevilacqua et al., 2012; Martin et al., 2001; Huang et al., 2015; Matsui et al., 2017) show this strategy can achieve better results. Super-resolution tasks are similar to geographic downscaling tasks, using SR deep learning methods to downscale the*

*geographic data can effectively avoid the problems encountered by former downscaling methods, such as high biases, regional sensitivity, high computational cost, etc. Deep learning methods use deep layers to bridge the relationship from low-resolution to high-resolution data and have robustness against the sensitivity. The computational cost is very low once the model has been trained, during the using step, and the deep learning method can nest a wide range easily. Shen et al. (2023) proposed a near-surface air temperature downscaling network SNCA-CLDASSD. In this model, Shen, et al. used two attention blocks to downscale the input data called Cross-Attention based on Light-CLDASSD. However, only near-surface air temperature is considered in this work and the network was built on CLDAS, which cannot cover long-term years. Liu et al. (2023) used the terrain to guide the deep learning network for the downscaling task called terrain-guided attention network (TGAN) in Southwest China. TGAN used the digital elevation model (DEM) to build high-resolution temperature (at 2 meters) results. The range of TGAN used begins in 2018 and TGAN cannot be used in the historical situation. Zhong et al. (2023) proposed a transformer-based learning method Uformer, which directly adds topography data, to achieve high-resolution meteorological variables in inner Mongolia province, China. Although topography data can help rebuild the high-resolution, directly adding into the input low-resolution will lose the characters of topography. All of the above, existing advanced deep learning methods of meteorological downscaling mostly used attention architecture (Transformer is one of the special attention architectures). However, now existing methods focus on one or two meteorological variables, while different variables have correlations and deep learning could handle multiple variables simultaneously. This way, not only saves the computational resources but also improves the performance of the model. Last and most important, there are no models that can cover a long-term and wide range of historical multiple variables."*

**Q4:** The authors are suggested to present the motivations of using the variables of temperature at 2m, pressure at the surface, and wind speed at 10m for GeoAN in this work.

**A4:** Thanks a lot for the suggestions. Nowadays, existing methods usually utilize W10m or T2m as the downscaling variables (Shen et al., 2023; Liu et al., 2023; Zhong et al., 2023;). We compared the variables between CLDAS and ERA5, and chose the shared variables in both datasets including temperature at 2m, pressure at the surface, wind speed at 10m, and precipitation. We actually used the precipitation in the network first, but there are several reasons we have to abandon this variable even if the precipitation is very valuable for use.

    **Point 1:** The evaluation of precipitation between CLDAS and ERA5 showed the low correlation is the main reason we discarded the precipitation and used the rest three variables. In future work, we will try to solve the problems of downscaling the precipitation.

    **Point 2:** The precipitation is a rapidly changing variable comparing the pressure or temperature. This led to deep learning or other statistical methods very hard to harvest the rules from the training data. Thus, the results will be not very good because of the irregular situation.

    **Point 3:** Precipitation is more like an extreme event, and deep learning is lean to produce an average result. Therefore, the results of the deep learning is hard to restore the value from the low-resolution input.

The related discussion can be found in the revised manuscript:

    *Lines 96-99 in the revised version:*

    *"There are four meteorological variables, temperature at 2m, pressure at the surface, wind speed*

*at 10m, and daily total precipitation (TP) considered in GeoAN. Considering it is hard to process the downscale of TP, only three other variables are produced by GeoAN in MDG625."*

*Lines 217-221 in the revised version:*

*"The dataset referenced in MDG625 includes three meteorological variables (PRS, T2M, and WS10m), but it lacks precipitation data. The low correlation of total precipitation (TP) between the ERA5 and CLDAS datasets is the primary reason why the GeoAN model struggles to accurately restore the spatial structure of precipitation. Inspired by the work of (Zou et al., 2024), more comprehensible algorithms could be implemented in GeoAN in the future"*

**Q5:** Page 6, line 98, "GeoAB is repeated 18 times", What considerations are there for setting it to 18?

**A5:** The repeated times is an adjustable hyperparameter. The reasons we set it to 18 are listed in the following discussion.

**Point 1:** Deep learning methods commonly use repeated blocks to reach a deep network for harvesting deep information, and that's why it is called "deep learning". In this theory, the deeper the network is, the more performance it will have. However, when the network is too deep, the chain rule will cause exploding or vanishing gradient problems. The computational expense of a deeper network is huge and the training step will last several weeks or months. Thus, the repeated times can not be unlimited deep, and it is constricted by the hardware.

**Point 2:** The data is the soul of the deep learning methods. More training data can produce better performance and emergence. This performance is based on a large model and enough data. If the data is not enough for the large model, it will damage the performance and the general practice is using a smaller model. Thus, the repeated times are not allowed too big.

We last chose 18 as our repeating times, and it served as the result of balancing the computational resources and the depth of the network. If we choose a deeper network, our GPUs (4 RTX 6000 Ada GPU 48G) can't afford the computational expense and may cause exploding or vanishing gradient problems. From another opinion, considering the training data, 18 is a reasonable setup. We add the above reasons of setting the repeated times to 18 in the caption in Fig. 1 in the manuscript.

*Caption in Fig. 1 in the revised version:*

*"The results of the two blocks of head in the diagram have the same channels of 108. GeoAB, which is repeated 18 times constricted by the hardware and the data amount, is the attention block for extracting deep information using geopotential."*

**Q6:** Page 6, line 105, "GUP memory limitation" should be "GPU memory limitation"?

**A6:** Sorry about this typo, and we thank you for your attention to the detail, it should be *"GPU memory limitation"*. We have revised this mistake in the manuscript and checked the whole paper to avoid the same mistake.

**Q7:** In addition to UNet and SwinIR, the authors are suggested to compared the results of the proposed method with more existing representative deep learning based downscaling methods.

**A7:** Your suggestion is insightful for us to improve this work. As to the questions in the discussion, here

are our replies. As far as we know, our work is the first attempt to establish the mapping relationship from ERA5 to CLDAS to make a long-term daily meteorological dataset. Thus, there are no existing works that can be compared directly. Downscaling is a similar task to super-resolution in computer vision, so we choose three super-resolution methods migrating to this task to make a comparison:

    1) Bilinear is the most classic and commonly used algorithm to improve resolution.

    2) U-net has been one of the most used deep learning methods in recent years in various tasks for its strong universality.

    3) SwinIR, proposed in 2021, was a novel SOTA (state-of-the-art) super-resolution model based on a transformer block. It can be considered the benchmark and representative deep learning method of SR-related missions.

To migrate the methods to downscale the long-term meteorological dataset from ERA5 to CLDAS, each existing method has to be redesigned. These methods may take several weeks to months to retrain to be compatible with the task. The above discussion can be found in a new section "Limitations" in the revised version:

*Line 221 in the revised version:*

"We will try to compare a wider range of models to evaluate our performance."

We redesign the deep learning networks for similar parameter quantities or GPU costs. We keep the hyperparameters for a fair comparison during the training and testing step. The details of the redesign are listed as follows:

*Lines 150-156 in the revised version:*

*"To ensure a fair comparison, the U-Net architecture was modified for the downscaling task, resulting in a customized version referred to as U-Net Evolution (U-Net Evol.). The original U-Net implementation is available at https://github.com/milesial/Pytorch-UNet, while the SwinIR code can be accessed at https://github.com/JingyunLiang/SwinIR. To maintain consistency, all deep learning models were configured with equivalent parameters or computational complexity, and they were trained for 100 epochs using identical hyperparameters under the same environmental conditions."*

In the future, we will continue to make larger comparisons with more existing representative deep learning methods and delve into how these deep learning methods can dig geographic information to help downscale the variables.

**Q8:** The proposed method has been used to generate a long-term dataset; however, the validation dataset in this paper is quite limited. It is suggested to evaluate the accuracy and reconstruction quality in terms of PSNR and SSIM using a larger dataset.

**A8:** Thank you so much for the insightful suggestions. More evaluation data could produce a more convincing result. We will try to do a more comprehensive comparison in future work. However, there are some considerations about how many years we should use to evaluate this work we have considered.

**Table AC1-1.** The data range of different deep learning downscaling methods is used.

|  | Training dataset | Testing dataset |
|---|---|---|
| Zhong et al. (2023) | 2020.01 – 2021.05 | 2021.06 – 2021.09 |
| Liu et al. (2023) | 2019.01 – 2019.12 | 2018 |
| Shen et al. (2023) | 90% of 2016, 2017, 2019, 2020 | 2018 |

**Point 1:** Due to many researchers (Zhong et al. (2023); Liu et al. (2023); Shen et al. (2023)), as shown in **Tab. AC1-1**, only using the data in one or two years for downscaling one or two meteorological parameters and one or fewer years for evaluation, we finally decided to use 3 years for training and 1 year to evaluate the model.

**Point 2:** The amount of the total data is fixed and if we use more data to evaluate the model, less data can be used in training. For a better performance, we use 3 years for training and 1 year for evaluation.

**Point 3:** The validation dataset needs to cover one full cycle, for the meteorological task is one year. Although there are slight distributing differences between different years, using a whole year of data for validation is enough to cover most situations.

**Point 4:** The performance of training the deep learning can be obtained from the training log (If the model has converged). The training log is shown in **Fig. AC1-1**. We used the pre-trained model to train the GeoAN and we can find the network convergence is relatively optimistic.

[Figure]

**Figure AC1-1.** The original training logs of GeoAN. The final GeoAN is trained from the pre-trained model: geoan-2024-0412-1935-4239 (recorded as version-4239), and the version-4239 is trained from version-4399.

A larger dataset and more compared methods are helpful to evaluate GeoAN and MDG625's quality. Limited by the hardware and the availability of data, the larger-scale evaluation has not been conducted yet. In the future, we are going to consider multimodal data (such as rain gauges, satellites, etc.) and use more types of algorithms for analysis to downscale more variables such as precipitation and draw a more comprehensive conclusion.

**References:**

Bevilacqua, M., Roumy, A., Guillemot, C., and Alberi-Morel, M. L.: Low-complexity single-image

super-resolution based on nonnegative neighbor embedding, in: The British Machine Vision Conference, pp. 1–10, 2012.

Di Luca, A., de Elía, R., and Laprise, R.: Challenges in the quest for added value of regional climate dynamical downscaling, Current Climate Change Reports, 1, 10–21, 2015.

Giorgi, F. and Gutowski Jr, W. J.: Regional dynamical downscaling and the CORDEX initiative, Annual review of environment and resources, 40, 467–490, 2015.

Huang, J.-B., Singh, A., and Ahuja, N.: Single image super-resolution from transformed self-exemplars, in: Proceedings of the IEEE Conference on Computer Vision and Pattern Recognition, pp. 5197–5206, 2015.

Liang, J., Cao, J., Sun, G., Zhang, K., Van Gool, L., and Timofte, R.: Swinir: Image restoration using swin transformer, in: Proceedings of the IEEE/CVF International Conference on Computer Vision, pp. 1833–1844, 2021.

Liu, G., Zhang, R., Hang, R., Ge, L., Shi, C., and Liu, Q.: Statistical downscaling of temperature distributions in southwest China by using terrain-guided attention network, IEEE Journal of Selected Topics in Applied Earth Observations and Remote Sensing, 16, 1678–1690, 2023.

Martin, D., Fowlkes, C., Tal, D., and Malik, J.: A database of human segmented natural images and its application to evaluating segmentation algorithms and measuring ecological statistics, in: Proceedings of the IEEE/CVF International Conference on Computer Vision, vol. 2, pp. 416–423, IEEE, 2001.

Matsui, Y., Ito, K., Aramaki, Y., Fujimoto, A., Ogawa, T., Yamasaki, T., and Aizawa, K.: Sketch-based manga retrieval using manga109 dataset, Multimedia Tools and Applications, 76, 21 811–21 838, 2017.

Song, Z. and Zhong, B.: A Lightweight Local-Global Attention Network for Single Image Super-Resolution, in: Proceedings of the Asian Conference on Computer Vision, pp. 4395–4410, 2022.

Shen, Z., Shi, C., Shen, R., Tie, R., and Ge, L.: Spatial Downscaling of Near-Surface Air Temperature Based on Deep Learning Cross-Attention Mechanism, Remote Sensing, 15, 5084, 2023.

Teutschbein, C. and Seibert, J.: Bias correction of regional climate model simulations for hydrological climate-change impact studies: Review and evaluation of different methods, Journal of hydrology, 456, 12–29, 2012.

Zhong, X., Du, F., Chen, L., Wang, Z., and Li, H.: Investigating transformer-based models for spatial downscaling and correcting biases of near-surface temperature and wind-speed forecasts, Quarterly Journal of the Royal Meteorological Society, 2023.

Zou, W., Hu, G., Wiersma, P., Yin, S., Xiao, Y., Mariethoz, G., and Peleg, N.: Multiple-point geostatistics-based spatial downscaling of heavy rainfall fields, Journal of Hydrology, 632, 130 899, 2024.